# Competitive binding of actin and SH3 domains at proline-rich regions of Las17/WASP regulates actin polymerisation
Lewis P. Hancock[1,2], John S. Palmer[1,2], Ellen G. Allwood[1], Iwona I. Smaczynska-de Rooij[1], Anthony J. Hodder[1], Michelle L. Rowe ®[1], Mike P. Williamson ®[1] & Kathryn R. Ayscough ®[1] ✉

Eukaryotic actin filaments bind factors that regulate their assembly and disassembly creating a self-organising system, the actin cytoskeleton. Despite extensive knowledge of signals that modulate actin organisation, significant gaps remain in our understanding of spatiotemporal regulation of de novo filament initiation. Yeast Las17/WASP is essential for actin polymerisation initiation supporting membrane invagination in *Saccharomyces cerevisiae* endocytosis and therefore its tight regulation is critical. The adaptor protein Sla1 inhibits Las17 but mechanisms underpinning Las17 activation remain elusive. Here we show that Las17 binding of tandem Sla1 SH3 domains is >100-fold stronger than single domains. Furthermore, SH3 domains directly compete with G-actin for binding in the Las17 polyproline region, thus rationalising how SH3 interactions can affect actin polymerisation despite their distance from C-terminal actin-binding and Arp2/3-interacting VCA domains. Our data and proposed model also highlight the likely importance of multiple weak interactions that together ensure spatial and temporal regulation of endocytosis.

Actin is a highly conserved protein functioning along with its associated proteins as a structural filamentous network facilitating cell organisation, membrane trafficking and cell motility. Actin filaments are double-stranded helical structures generated from actin monomers (G-actin) in a process termed nucleation. Initial nucleation is energetically unfavourable and in cells requires the action of nucleating proteins which, through different mechanisms, facilitate recruitment and organisation of the monomers to allow polymerisation. There are several classes of actin nucleators including the seven-membered Arp2/3-complex which leads to generation of branched actin networks, while others are simpler linear organisations of actin binding motifs on single polypeptides[1–4].

WASP proteins have been recognised to play a critical role in actin nucleation, and this has been largely attributed to their C-terminal VCA region which can bind both to an actin monomer through its WH2 domain, and also to the Arp2/3 complex, thus activating Arp2/3-mediated nucleation[5]. This has led to WASP family proteins often being referred to as nucleation promotion factors. Upstream regions of WASP family proteins including their polyproline regions (PPR) have been proposed to localize and potentially regulate WASP activity[6,7].

Budding yeast, *Saccharomyces cerevisiae*, has been highly informative and tractable for studies of actin regulation in cells. During endocytosis >50 different proteins assemble at cortical sites to facilitate plasma membrane invagination and scission. Whilst endocytic coat proteins can reside for minutes at the membrane, recruitment of a complex comprised of the single yeast WASP family protein Las17/Bee1 and its negative regulator Sla1, triggers a relatively precise sequence of events lasting about 30 s culminating in invagination and vesicle scission[8–11]. Las17/WASP has a similar domain structure to WASP-family proteins in other eukaryotes and promotes Arp2/3 activity[12,13]. Arp2/3 itself arrives at endocytic sites about 10–15 s after Las17, but the origin of the filaments to which it must bind to nucleate further filaments remains debated.

Studies from both yeast and other cell types demonstrate that the PPR of WASP-family proteins can play a more integral part in actin polymerisation than originally considered. In yeast, deletion of the actin and Arp2/3 binding VCA domain of Las17 delays but does not prevent actin polymerisation or endocytosis whereas the PPR is critical[11,14]. The Las17 PPR itself has been shown to directly bind actin, and in vitro the N terminal part of the region (PPR-N) can promote actin filament formation while the C-terminal PPR (PPR-C) appears to stabilise F-actin[15–17]. Most recently, in both a mammalian B16-F1 cell line and in *Dictyostelium* cells, the PPR of Scar/WAVE was demonstrated to be essential for actin polymerisation, while the VCA domain was required for generating a branched actin network. In cells expressing Scar/WAVE lacking the VCA region, actin protrusions were still generated indicating an Arp2/3-independent mechanism

[1]School of Biosciences, University of Sheffield, Sheffield, UK. [2]These authors contributed equally: Lewis P. Hancock, John S. Palmer.
✉e-mail: k.ayscough@sheffield.ac.uk

functioning at the cell periphery[18]. A key question emerging from these studies is therefore how the WASP-family protein PPR functions in actin polymerisation. The actin binding protein profilin is known both to bind the PPR in some WASP family proteins and to enhance actin polymerisation[19]. However, our previous studies demonstrate that profilin does not bind the PPR of Las17 nor is it required for the polymerisation activity of Las17/ WASP[15].

A further question that emerges is how WASP family facilitation of actin polymerisation is regulated. There are again insights from yeast. The multi-domain protein Sla1 has three N-terminal SH3 domains that bind to Las17, as well as other domains that interact with clathrin and endocytic cargoes[20–23]. In biochemical actin polymerisation assays, the SH3 domains of Sla1 inhibit the ability of Las17 to activate nucleation by Arp2/3[13]. It has remained unclear however, why binding of Sla1 SH3 domains to the PPR-N should affect actin nucleating function at the VCA region more than 100 amino acids away especially as this intervening region is largely unstructured.

Another important aspect of a regulatory interaction is a mechanism to switch between inactive and active forms of proteins. Central to this is also a requirement that alleviation only occurs at the appropriate time and location in cells. In yeast, it was demonstrated that the small GTPase Sec4 can alleviate Sla1 inhibition of Arp2/3-Las17-mediated actin nucleation. This demonstrated a direct link from cargo to activation of actin polymerisation for endocytosis, though the mechanism underpinning alleviation of inhibition is not known[24].

In the work presented here we report the structure of tandem Sla1 SH3 domains binding to Las17; the binding of up to three actin monomers to Las17 PPR; and provide evidence for mechanisms that could contribute to alleviation of Sla1 inhibition of Las17 activity.

## Results

### The importance of tandem SH3-binding for inhibition of Las17

To increase understanding of the mechanism of Las17 inhibition mediated by Sla1 SH3-binding we first investigated the role of single and joined SH3 domains. In Sla1 the first two SH3 domains are immediately juxtaposed with no linker. There are then 236 residues before the third SH3 domain (Fig. 1A). The in vivo importance of these domains has been demonstrated, including their order[11,20,25]. In vitro, Sla1 SH3 domains 1, 2 and 3 have been added singly and conjoined to actin polymerisation assays in the presence of Arp2/3 and Las17[13]. These experiments revealed that SH3 domains 1 and 2 but not 3, inhibited about 80% of actin polymerisation activity. Inhibition was slightly increased (to about 85%) when all three SH3 domains were joined. These published studies used 15 nM Las17 and a marked excess (1 μM) of Sla1 SH3 domains. Since then proteomic studies have reported protein abundance in yeast cells, and these data have been unified into one dataset suggesting Las17 has a median abundance of 2743 ± 1007 molecules per cell and Sla1 8674 ± 2996 molecules per cell giving an approximately 1.5–3-fold excess of Sla1 over Las17[26]. In this work we have used around equimolar protein concentrations to investigate Sla1 inhibition of Las17 closer to physiological ratios. We first sought to determine whether inhibition of actin polymerisation by Sla1 SH3 domains was observed in the presence of Arp2/3 and Las17 using a 1:1 ratio of SH3 domain to Las17. As shown in Fig. 1B, while substantial inhibition of actin polymerisation is observed when a peptide carrying all three Sla-SH3 domains is added, single SH3 domains were relatively ineffective in inhibiting Las17-Arp2/3 activity. This suggests that binding avidity of SH3 domains is likely to be relevant for inhibition in cells.

To investigate binding characteristics of the SH3 domains further, Sla1 SH3 protein domains were purified. Three Sla1 SH3 single domains, tandem domains Sla1 SH3#1&2, Sla1 SH3#1&2 W108A (in which the W108A mutation prevents proline motif binding by SH3#2 but not SH3#1) and a fragment with all three SH3 domains were expressed recombinantly and purified. Binding affinity for the Las17 PPR-N fragment (residues 300–422) was then measured using Biolayer Interferometry (BLI) as described. Affinities were obtained for all domains except for Sla1 SH3#2 alone which

showed no binding. We also tested binding of Sla1 SH3#2 using microscale thermophoresis (MST) and while weak binding was detected, it could not be saturated within the concentration range. Binding curves and affinities are shown in Fig. 1C. The data reveal binding affinities of individual domains (#1 and #3) of 7.5 μM and 16 μM respectively. In addition, binding of tandem domains Sla1 SH3 1&2-W108A bound with similar affinity (7.4 μM) to #1 alone. However, when both Sla1 SH3#1&2 domains were competent for binding this affinity increased almost 100-fold to 87 nM. There was then a smaller contribution when SH3#3 is also linked, with an apparent $K_d$ of 39 nM for the three domains. This outcome is in line with original research of Rodal and colleagues and with more recent studies of avidity developed by Williamson, which highlighted the importance of tandem binding SH3 domains[13,27].

## Structure of Sla1 SH3 domains and binding to Las17 polyproline peptides

We next sought to determine how the arrangement and structure of the N-terminal tandem SH3 domains of Sla1 influence its binding to Las17. We used Alphafold2.0 to generate a predicted structure of Sla1 SH3#1&2[28,29]. We focussed on the two N-terminal domains as these conferred most binding to Las17. The most favoured structure (Fig. 2A) showed the two SH3 domains with several interdomain hydrogen bonds and with their proline helix binding faces at 180° to one another. Based on modelling, polyproline motifs would need to be at least 28 amino acids apart to bind two SH3 domains.

To investigate further, we purified $^{15}N$, $^{13}C$ labelled Sla1 SH3#1&2 and analysed using NMR. A full assignment was obtained (Fig. 2B) which confirmed the Alphafold structure. An important aspect when considering the function of tandem domains is the level of rigidity or flexibility between the domains. The structure predicts hydrogen bonds between the SH3 domains suggesting they could act as a single domain rather than as two independent domains. NMR was used to determine relaxation parameters ($^{15}N$ $R_1$, $R_2$ and NOE) of $^{15}N$-labelled SH3#1-2 (Fig. 2B). These results show localised increased mobility for the loop around residues D52-E55 in SH3#1 (indicated by the reduced values for $R_2$ and NOE in the region indicated by a purple bar in Fig. 2C) but indicate that residues in the linker between the two SH3 domains (residues E67-V70) have the same correlation time as most other residues in the protein (ie, no reduction in $R_2$ and NOE in the region indicated by a red bar in Fig. 2C), thus demonstrating that the two SH3 domains behave as a single rigid unit. The structural unity of the two-domain structure explains why attempts to study SH3#2 alone were unsuccessful.

To gain further insight into the interaction of Sla1 SH3 domains with Las17, Sla1 SH3#1&2 and Sla1 SH3#3 were $^{15}N$-labelled and purified. NMR was used to measure the relative binding affinities of peptides corresponding to the three proline motif regions in the Las17 N-terminal half of the PPR (see Methods). This 300-422 fragment has previously been shown to contain the proline motifs responsible for most SH3 domain binding interactions (Fig. 3A)[30]. The NMR results (Fig. 3B, C, D) show that all three PP peptides induce chemical shift changes throughout the protein, and thus bind to the tandem SH3#1&2 domain. The affinity of the peptides for the first SH3 domain is approximately eight times stronger than for the second (e.g. for peptide 3, a $K_d$ of 24 μM vs 190 μM, shown by the greater curvature on the fitted values in the top panel of Fig. 3C than in the bottom panel). All three peptides have a similar pattern of shift changes (Fig. 3D) and therefore bind to the same location on the two domains (Fig. 3E), which is centred on the surface-exposed tryptophan residues. Peptide PP1 binds approximately 4 times more weakly than peptides PP2 and PP3. All three peptides bind considerably more weakly to SH3#3 (Supplementary Fig. 1), with all three peptides having similar affinities, and binding in the same location, which again includes the exposed tryptophans. These data, together with the 28 amino acid residue spacing required between motifs to bind the conjoined SH3 domains 1&2 of Sla1 (based on the modelling discussed above), suggest that Sla1 SH3#1 is most likely to bind PP2 or PP3, SH3#2

**Fig. 1 | SH3 domains of Sla1 bind Las17 and inhibit Arp2/3-dependent nucleation of actin polymerisation by Las17. A** Schematic diagram outlining one possible arrangement of Sla1 SH3 domains bound to the N-terminal PPR of Las17 (amino acids 300–422). SH3 domains of Sla1 are numbered #1, #2 and #3. The PPR-N of Las17 is the region shown to bind SH3 domains. PP in pink boxes indicate poly-proline tracts in this region. Other regions of Las17 are included but not to scale. **B** Representative pyrene actin assays show inhibition of Las17-Arp2/3 mediated actin polymerisation by Sla1-SH3 domains when added as separate domains (orange #1, green #2 or cyan #3) or on a single peptide (red). Actin; 3 µM, Las17; 300 nM, and Sla1 SH3 domains; 300 nM each. **C** Biolayer Interferometry measurements of the affinity of Sla1 SH3 domains for Las17 (aa 300–422). Sla1 SH3#1 (aa 3–68), SH3#3 (aa 354–413), SH3#1-2 (aa 5–131), SH3#1-3 (aa 5–413).

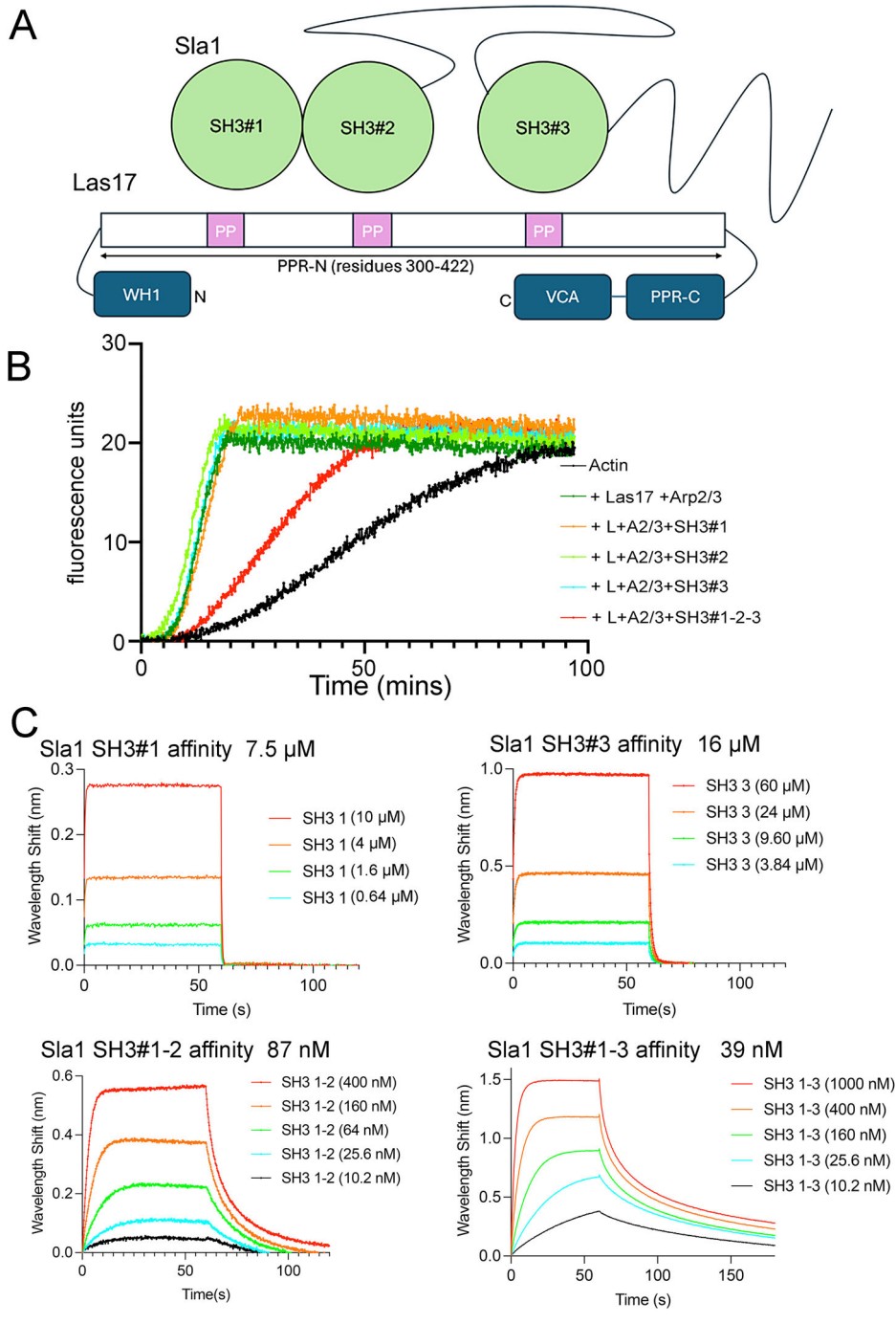

Actin binds to three sites in the PPR of Las17

The next important question we aimed to address was why SH3 binding in the Las17 PPR impacts actin polymerisation. As found with a subset of other WASP family proteins, the C-terminal VCA domain of Las17 is not alone sufficient for effective activation of Arp2/3-mediated actin polymerisation[13,31]. We and others have shown that monomeric actin can bind to the Las17(300-422) PPR region of Las17[15–17]. To determine further how Las17 PPR is affecting actin polymerisation we focussed on a small region of Las17 encompassing two sites including arginine pairs previously shown to be involved in G-actin binding, Las17(342–392), Fig. 3A[15,16]. Using microscale thermophoresis (MST) a binding affinity ($K_d$) of

780 ± 170 nM was calculated for the Las17 fragment to G-actin (Supplementary Fig. 2). This is about 10-fold weaker than that found for the longer fragment (amino acids 300–422) under similar conditions, suggesting that additional G-actin binding capacity might lie outside this previously identified region[15]. In pyrene actin polymerisation assays this Las17 peptide was unable to stimulate Arp2/3-independent actin polymerisation, although a longer fragment, Las17(300–633) containing both PPR and VCA, was able to stimulate actin polymerisation in the absence of Arp2/3 (Fig. 4A).

These outcomes, coupled with the residual actin binding of Las17 fragments carrying mutagenised arginines (RR349,350AA and RR382,383 AA)[15,16] led us to consider that there was a high likelihood of a further actin binding site in the Las17(300-422) fragment. We noted that in the identified G-actin binding sites, arginines were adjacent to a 5×proline tract in a sequence RRGPAPPPPP. While not

would bind whichever of these two was not bound by SH3#1 and then SH3#3 would most likely bind PP1.

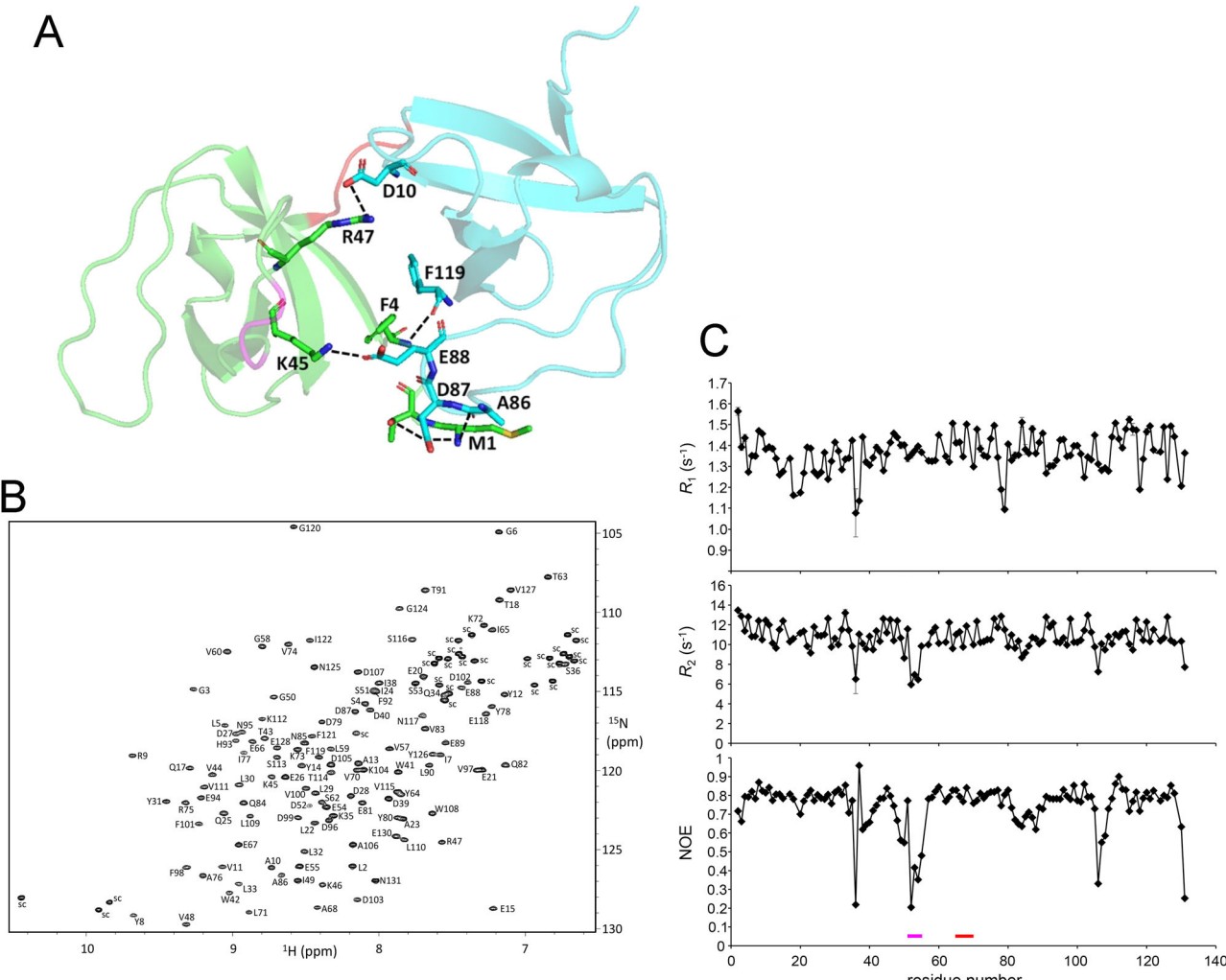

**Fig. 2 | A structural analysis of Sla1 SH3 domains 1 and 2. A** The predicted interface between the first two SH3 domains of Sla1 as modelled using Alphafold. SH3#1 is shown in light green and SH3#2 is shown in cyan. Residues predicted to form hydrogen bonds using the Pymol search function are shown as sticks and labelled with their respective residue numbers. Dashed black lines indicate interactions. The red line in the structure indicates residues 67–70 in the hinge between the SH3 domains; the magenta loop indicates residues 52–54. **B** Assignment of the

$^1$H $^{15}$N HSQC spectrum of Sla1 SH3 domains 1 and 2. **C** $^{15}$N relaxation data for Sla1 SH3#1&2. The linker between domains is approximately residues 67–70 (shown with red line under the trace and in **A**), and the mobile region 52-54 is part of a loop (shown with magenta line below the trace and in **A**). The correlation time of 8.5 ns (derived from the trimmed $^{15}$N $T_1/T_2$ ratio), matches that calculated for a rigid two-domain structure rather than two independent domains (8.8 and 4.5 ns, respectively).

an exact match, a third possible motif was noted at Las17 residues 319–328 R̲NNR̲PVPPPPP. Mutations were generated at arginines 319 and 322 in a plasmid already carrying the other paired arginine mutations. MST was used to measure binding to labelled actin as described. The G-actin binding sites are numbered to align with the PP motifs in Fig. 3A, i.e. ABS1 being tested is overlapping with PP1; ABS2 with PP2 and PP3 with ABS3. As shown in Fig. 4B, mutation of ABS1 (R319A, R322A) in combination with ABS3 alone or with both ABS2 (R349A, R350A) and ABS3 (R382A, R383A) causes marked reductions in binding affinity of actin for Las17. Calculation of binding affinity for G-actin from the MST data demonstrates that in G-buffer wild-type Las17 300-422 had a $K_d$ 90.4 ± 4.2 nM. Mutation of ABS2 (RR349, 350AA) had reduced affinity of 191 ± 7.4 nM. Mutation of ABS3 (RR382,383AA) reduced affinity to 119 ± 6.2 nM. Mutation ABS1Δ ABSΔ3 had further reduced affinity of 546 ± 4.7 nM. The fragment with all arginine pairs mutagenized (ABSΔ1Δ2Δ3) had a $K_d$ of >7410 ± 820 nM demonstrating that each paired mutation causes a reduction in binding to G-actin and supports the designation of three distinct actin binding sites (ABS1, ABS2, and ABS3) overlapping the PP motifs 1, 2 and 3 respectively.

## A structural model of Las17 binding G-actin

To ascertain how these three G-actin binding sites could be pivotal in the regulation of actin polymerisation we asked how actin monomers bind the PRR, and whether it would be possible for multiple actin monomers to bind at one time. For this we used structural modelling approaches, HPEPDOCK2.0[32,33] (Fig. 5A–C; Supplementary Fig. 3A, B) and PIPER-FlexPepDock[34] (Supplementary Fig. 3C). Using either approach, the majority of predictions showed docking of the three Las17 ABS peptides at the barbed end groove in G-actin. For PP3/ABS3 (Las17 residues 377-396), the paired arginines were most often oriented towards the 'front' face of the actin monomer (Fig. 5A) and there were interactions with actin E334 (Fig. 5B). It was also noted that in this arrangement, aromatic rings of actin (Y143, F169, and F375) would lie in relatively close proximity to the consecutive proline residues of the peptide, as is typically seen in protein interactions with PPRs[35].

With the other peptides, Las17 319-340 (PP/ABS1) and Las17 330-362 (PP/ABS2), while docking was at the barbed end groove (Supplementary Fig. 3 A, B), the docking sites showed some variability in their arginine interactions. Each modelled interaction site involved three arginines (two N-terminal to the polyproline tract and one C-terminal). Together the data

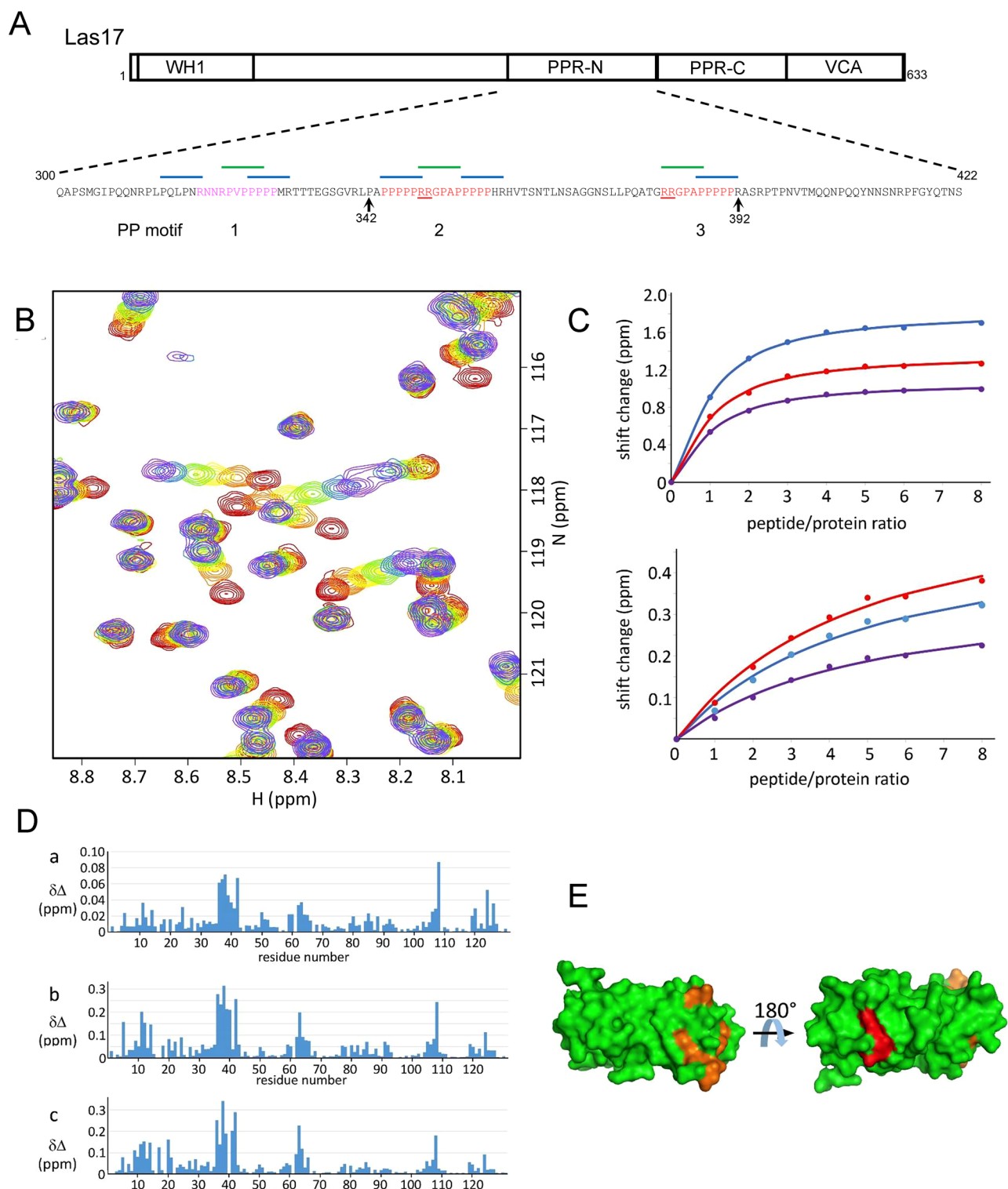

indicate two predominant arginine-interacting sites flanking the hydrophobic barbed-end groove. These include residue E334 at the front face of actin, and E361/E364 at the rear face (Fig. 5C).

Having docked the single peptides and actin monomer, we then considered the spacing of the PP motifs/actin binding sites on Las17 and whether binding of three monomers is sterically achievable. Figure 5D shows a model indicating a possible arrangement of three actin monomers along the length of the Las17 fragment. The spacing of the actin binding sites identified is reminiscent of that found with actin interactions with the

tandem WH2 domains of Spire and other longitudinal actin nucleators including JMY[2].

## Las17-activated actin polymerisation in the presence and absence of Arp2/3

Previously we revealed that, while not an absolute requirement, the presence of the WH2 domain at the C-terminus of Las17 could enhance actin polymerisation even in the absence of Arp2/3[15]. It is possible that the WH2 binds G-actin and delivers it laterally to the longitudinally

**Fig. 3 | NMR titrations of Sla1 SH3 domains 1 and 2 with Las17 peptides.**
**A** Schematic diagram of Las17. The PPR from residues 300-535 is subdivided into PPR-N (300-422) and PPR-C (423-535). The amino acid sequence is shown for PPR-N with three polyproline (PP) motifs indicated. Lines above the sequence are SH3 domain binding consensus sequences with green indicating type-I SH3 domain binding motifs, and blue type-II SH3 domain binding motifs. Underscored arginine residues in red text motifs have previously been identified as important in G-actin binding. Pink text indicates a predicted new ABS region. **B** A region of the HSQC spectrum from the titration of [15]N-labelled Sla1 SH3 domains 1 and 2 with peptide 3 is shown. For clarity only the titrations with 0, 1, 2, 3, 5 and 8 peptide equivalents are shown (red, orange, yellow, green, blue, purple respectively). **C** Chemical shift changes in protein signals were fitted to standard equations (Williamson 2013) to estimate binding affinity. The estimated affinities clearly fell into two groups: signals from SH3#1 (top panel) fitted to a common affinity of 24 μM, while signals from SH3#2 (bottom panel) fitted to a common affinity of 190 μM. The signals shown are (top) [15]N shifts for I38 (blue), W42 (red) and W41 (purple) (bottom) [1]H shifts for W108 (blue), G124 (red) and N85 (purple). **D** Chemical shift changes on addition of each peptide (PP1, PP2, PP3) are shown as the weighted chemical shift changes for [1]H and [15]N, $[\Delta\delta_H^2 + (0.14\Delta\delta_N^2)]^{1/2}$. The three peptides bind at similar locations, although peptide 1 binds approximately three times more weakly. The approximate affinities for PP1, PP2 and PP3 obtained from these data are respectively 70, 22 and 24 μM for SH3#1 and 550, 160 and 190 μM for SH3#2. **E** The binding site on the protein for peptide 3. Chemical shift changes on addition of peptide 3 were used to calculate the mean and standard deviation weighted shift change for each protein residue. Residues with shift changes larger than (mean + sd) are indicated in orange for domain 1 and red for domain 2, and comprise V11, Y12, Y14, S36, I38, D39, W41 and W42 (domain 1), and D105, A106 and W108 (domain 2). The figure shows two views rotated by 180°.

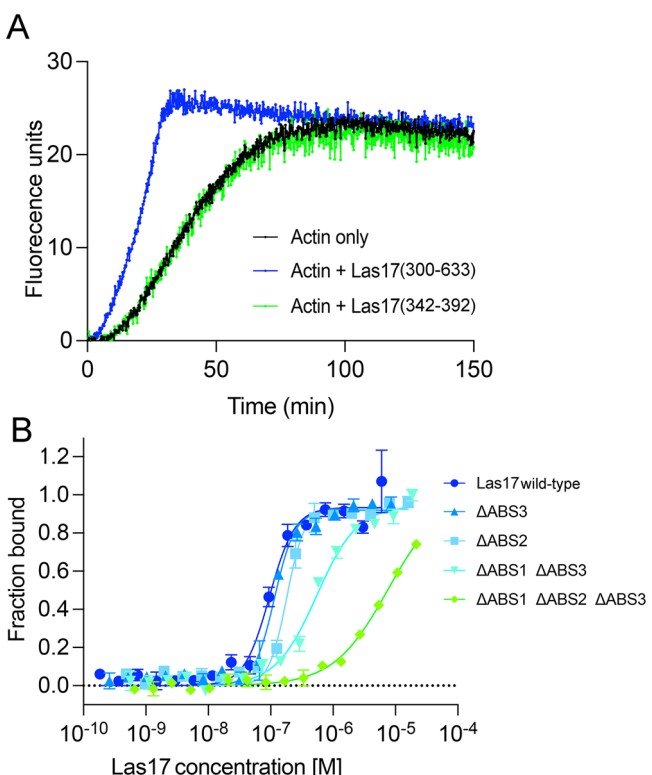

**Fig. 4 | Three sites in Las17 PPR bind to actin. A** A representative pyrene assay comparing the impact of Las17 fragment (300–633) with the minimal fragment (342–392) on actin polymerisation (in the absence of Arp2/3). 300 nM each fragment; 3 μM actin. **B** MST was used to measure binding of different concentrations of Las17 (wild type or with mutations in the actin binding sites (ABS)) to labelled actin. Error bars are standard error of mean. Green trace shows the impacts of all three sites mutagenized; dark blue is with no sites mutagenized.

bound monomers on Las17 thus initiating a second sub-filament as in the protein Cobl[2,36]. To assess how filaments generated in the presence or absence of Arp2/3 differ, a TIRF-based assay was used. Videos of representative assays are shown in Supplementary Videos 1, 2, 3 and combined in Supplementary Video 4. Image stills taken at the 900 second time point in the videos (time lapse frame 30) are shown in Figure 6A[37]. The effect of the Las17 fragment on the nucleation of actin filaments was measured using methodology described by Graziano and colleagues (Supplementary Fig. 4A,B)[37]. This analysis revealed that the Las17 fragment increased the nucleation rate two-three fold compared to actin alone. For example, at the 10 minute time point the number of filaments increased from 16.3 ± 2.3 for actin alone to 42.3 ± 7.0 for actin + Las17 fragment (errors are standard deviation).

The filaments formed in the presence of Las17 alone or with Las17 and Arp2/3 are qualitatively different. The filaments with Las17 alone are long and unbranched whilst in the presence of Arp2/3 there are more, short, branched filaments.

The growth rate of filaments was measured and as shown (Fig. 6B), addition of Las17 is sufficient to increase the elongation rate of actin filaments. Interestingly, addition of Arp2/3 does not increase the rate of elongation beyond that of Las17 alone indicating that the markedly different rates of polymerisation observed in pyrene assays is more likely due to the increase in the number of filaments from nucleating action of Arp2/3 on existing filaments[15].

### Factors affecting Las17-actin binding

Having established that both actin and Sla1-SH3 domains can bind Las17 at sites in the N-terminal PPR, we next determined whether binding occurs concurrently or whether SH3 domains directly compete for binding. Competition for PPR binding sites could be at least part of the mechanism underlying Sla1 inhibition of actin polymerisation. Relevant proteins and fragments were purified. G-actin binding to Las17 was established using MST, with an actin concentration of 50 nM, and Las17(300–422) concentrations between 0.29 and 9.5 μM. The experiment was then repeated with the addition of Sla1 SH3#1-3 fragment at 4.5 μM. As shown in Fig. 7A addition of Sla1 SH3#1-3 reduces the binding affinity of actin for Las17 with a 40-fold change in apparent $K_d$ indicating competition. This suggests a mechanism, whereby Las17/WASP-mediated actin polymerisation is regulated by competing actin and SH3-domain binding.

Given the importance of only generating actin filaments at appropriate sites in cells we postulated that various factors co-operate in releasing Sla1 from Las17 to facilitate actin filament polymerisation. Proteins reported to bind Sla1 N-terminal region which are also in an appropriate location include ubiquitin which binds Sla1-SH3#3, and the proteins Sla2 and Ysc84 which bind to the inter-SH3 domain region[10,38,39]. It was therefore of interest in the Alphafold analysis of Sla1 that an additional folded region was predicted in this area. Such a domain has not been reported in literature or in domain identification sites based on primary sequence analysis (Supplementary Fig. 5). We took the primary sequence of this domain (Sla1 253–339) and used threading software Phyre to identify domains with similar folds[40]. This identified a putative PH-domain. As PH-domains have been demonstrated to bind lipids[41], we purified Sla1 SH3#1-3 fragment containing the predicted PH domain and incubated with liposomes prior to sedimentation. As shown in Fig. 7B, C, only in the presence of liposomes does the fragment become enriched in the pellet fraction, demonstrating Sla1 1–413 is likely to bind membranes. We investigated the effect of liposome addition to Sla1-inhibited actin polymerisation assays but were unable to obtain a reproducible alleviation of Las17 activity demonstrating that binding of the Sla1 PH-domain, at least in our standard assay conditions, is not alone sufficient to relieve Las17 inhibition (Supplementary Fig. 5C).

A previous study demonstrated that after vesicle exocytosis, the small GTPase Sec4 binds Las17 in a complex with both Sla1 and another endocytic

**Fig. 5 | Modelling the Las17-actin interactions through global docking.** All residues are shown as a cartoon ribbon except for arginines that flank the polyproline sequence of the ABS sites, and key interacting actin residues. Yeast G-actin is shown in green (PDB: 1YAG). **A** Structures for four of the top ten HPEPDOCK predictions of ABS3 docking show interactions in the barbed end groove of the actin monomer. Peptides are shown in different colours. **B** All four of these structures coordinate actin residue E334 via the double arginine pair (R6 and R7 in the docked peptide). **C** A structure illustrating one of the top ten structure predictions for ABS1 shows how arginines at each end of the peptide of Las17 may interact simultaneously with acidic residues flanking the barbed end groove of actin (E334, and E361/364). **D** All three docked peptide structures can be modelled to illustrate binding of three actin monomers along the Las17 peptide. The ABS sequences are shown in red whilst the adjoining sequences that were part of the modelling in pink. The dashed pink line indicates parts of Las17 with no structural information.

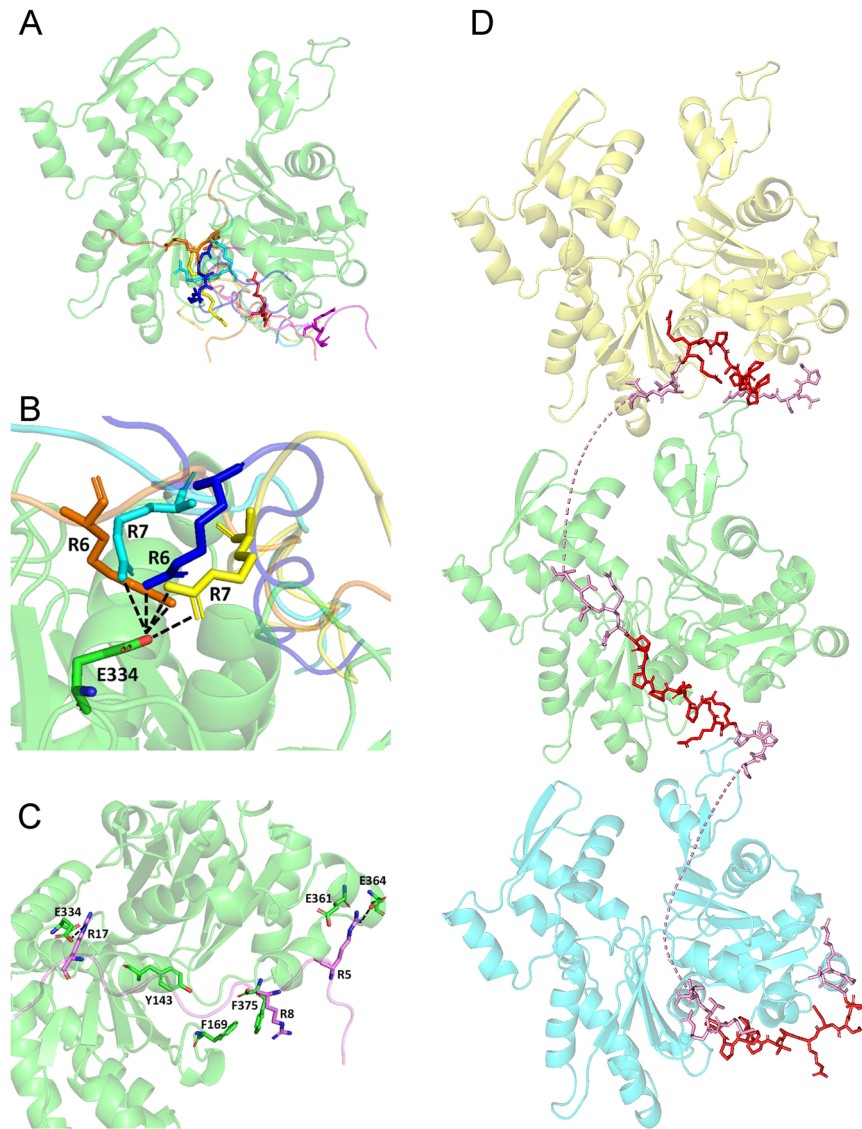

patch component Sla2[24]. In biochemical assays Sec4 alleviated Sla1-SH3 domain inhibition of Las17-Arp2/3-mediated actin nucleation. We therefore wanted to determine whether Sec4 could also alleviate inhibition of Arp2/3-independent actin polymerisation by Las17 as Arp2/3 is not found at endocytic sites at the earliest stage of cargo-recognition and binding. As shown in vitro in Fig. 7D, the addition of activated Sec4 to pyrene assays does indeed relieve inhibition of Las17-mediated actin polymerisation.

Earlier studies also showed Sec4 could bind directly to Las17 but not Sla1 in cell free extracts[24]. We used Alphafold2.0 to investigate sites of possible interaction between Las17 and Sec4 using full length Sec4 and the Las17 fragment 300-422. The predicted structure showed an interaction of Sec4 with a region of Las17 primarily between 327 and 333 which lies at the C-terminal end of the first PP motif (Fig. 3A). The model predicts an extensive interaction surface highlighted in red (Fig. 7E). Based on Pymol analysis predicted interactions include hydrogen bonds involving Las17 M330, R331, T332 and T333 as well as hydrophobic interactions potentially involving proline residues P327 and P328. Sec4 residues involved in the interactions are D56, F57, K58, I59, Q72, W74, and Y89. The overlap of this Sec4 binding site with the PP1 motif suggests that Sec4 binding could compete for SH3 binding at this motif thus allowing access for at least one actin monomer to bind in equilibrium.

### Reduced SH3 binding in an Sla1 P387A mutant

Our data, and that of others, suggest that when the Sla1/Las17 complex is recruited to endocytic sites, Sec4 and potentially other factors could act to weaken the Sla1 SH3 binding to Las17 and allow G-actin binding and filaments to be nucleated. To test if this idea was physiologically relevant we considered whether SH3 binding could be reduced while maintaining the actin polymerising function of Las17.

First, we carried out a spot-based peptide array analysis to determine sites of binding of GST-Sla1 SH3 domains within Las17. The peptide array carried 12mer peptides of Las17 with 2-residue offset (see Supplementary file for details). As shown in Fig. 8A Sla1-SH3#1 showed a preference for sequences in the Las17 proline tracts 2 and 3 (in agreement with our NMR data). SH3#2 only showed weak binding to spots, but again this was greater for tracts 2 and 3. In contrast, SH3#3 showed strongest binding for peptides in proline tracts 1 and 3.

Based on data from the actin-Las17 docking prediction and known SH3 proline interactions we considered that mutation of prolines might affect SH3 binding more strongly than actin[30,42]. The PP2 arginine pair appeared to be the strongest motif for actin binding (Fig. 4) so we also avoided mutating that region and focused on P387 and P388 as possible interaction residues in the third PP motif. In the binding assay (Fig. 8A) additional peptides were generated which included P387A and P388A

**Fig. 6 | Las17 impact on actin filament growth rate in the presence and absence of Arp2/3. A** Filament growth was followed using TIRF microscopy. Shown are stills from movies at 900 s (lapse rate 30 s in Supplementary videos). **B** Filament growth was measured as described in four independent experiments (data indicated by different colour spots on graph). Error bars are standard deviation. Ordinary one-way ANOVA statistical analysis. **** indicates p value < 0.0001.

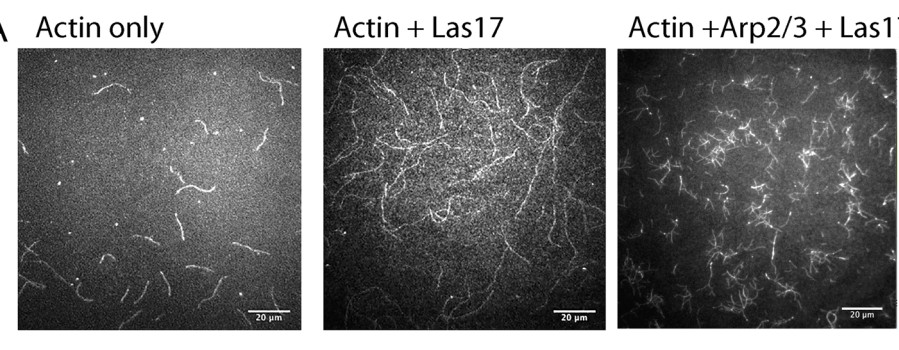

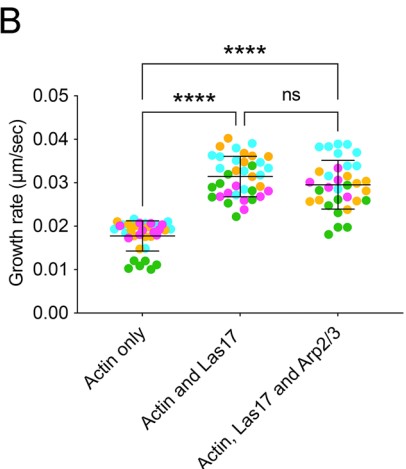

mutated versions of the strongest interacting peptide in PP3 (blue box in PP3). As shown, the mutation P387A (red box) but not P388A (green box) significantly reduced binding of all the Sla1 SH3 domains.

We then considered whether the P387A or P388A mutation in Las17 also affected G-actin binding. Constructs were made to express the Las17 (300–422) region with P387A and P388A mutations. MST analysis of actin to Las17 binding was carried out. As shown in Fig. 8B, there appeared little change in actin binding. The small reduction was not found to be statistically significant over three repeats.

### The effect of las17 P387A mutation in vivo

Having identified a Las17 mutation that reduced binding of Sla1 SH3 domains but not of actin in vitro, we sought to determine the impact of this in cells. Mutant las17 P387A would still be likely to bind Sla1 SH3 domains at the other two PP motifs but we thought the mutation could shift the actin-binding equilibrium, such that Las17 could be primed with G-actin prematurely before localisation to endocytic sites. Given the complexity of interactions involved in triggering and terminating endocytic invagination, whether this change would manifest as more rapid actin recruitment and invagination, or more chaotic, unregulated events was unknown.

A yeast strain was generated with an integrated mutation *las17 P387A* in place of the wild type *LAS17* gene. Expression of *las17 P387A* as the sole Las17 in cells was able to support growth across a range of temperatures but caused an increase in sensitivity to latrunculin-A (an actin monomer binding drug) (Supplementary Fig. 6A,B)[43]. Organisation of actin in cells was less well polarised and bulk phase endocytosis was slower than in wild type cells (Supplementary Fig. 6C,D). To gain further insight into the impact of the *las17 P387A* mutation, the behaviour of specific endocytic reporter proteins was analysed. The reporters were tagged with fluorescent markers in strains with the integrated *las17 P387A* variant. The effect of the mutation was determined for Las17-GFP itself, Abp1-GFP as an indirect marker of F-actin at the site and Arc15-mCherry to indicate recruitment of the Arp2/3 complex. Lifetime of the reporters at endocytic sites was measured for each

and the behaviour of individual fluorescent spots was analysed in the form of a kymograph which follows the assembly and movement of reporters over time (Fig. 9A, B, C). In terms of behaviour, wild type Las17 itself remains at the plasma membrane while other proteins such as Arp2/3, Sla1 and F-actin binding proteins show inward movement (invagination). Las17 P387A-GFP remained at the plasma membrane similar to its wild-type counterpart but its lifetime was nearly doubled. In the mutant background Abp1 has an increased lifetime and also has more aberrant behaviour including retraction of patches (Fig. 9B arrows). Arc15 also showed an increased lifetime compared to its behaviour in wild type cells.

Analysis of the three reporters indicates that reduced regulation of actin binding by Las17 causes aberrant events rather than simply enhancing the rate of endocytosis.

To understand relative changes in recruitment of proteins to endocytic sites we then analysed the Arc15 relative to Las17 itself. Strains were generated co-expressing Las17-GFP or las17 P387A-GFP with Arc15-mCherry. Individual spots were analysed for lifetimes and behaviours. As shown in the montages (Fig. 9D, E) in an otherwise wild-type strain background Las17 is recruited at the site and remains for about 22 s before Arc15 is recruited. There is then co-localization with dual-labelling for about 15 seconds, followed by a very short duration when Arc15-mCherry was observed alone (about 4 s). In the presence of las17 P387A, not only was the time prior to recruitment of Arc15 significantly longer (about 40 s) but also the duration of co-localization (28 seconds), and of Arc15 remaining recruited to endocytic sites following disassembly of Las17 (15 s). Together these in vivo data indicate that the *las17 P387A* mutation impacts all stages of endocytosis from its recruitment to final disassembly.

### Discussion

In this study we sought to gain insight into the function of the PPR region of a WASP family protein, how the region can contribute mechanistically to actin polymerisation and how this activity can be regulated positively and negatively. We have used the well-characterised context of actin-dependent

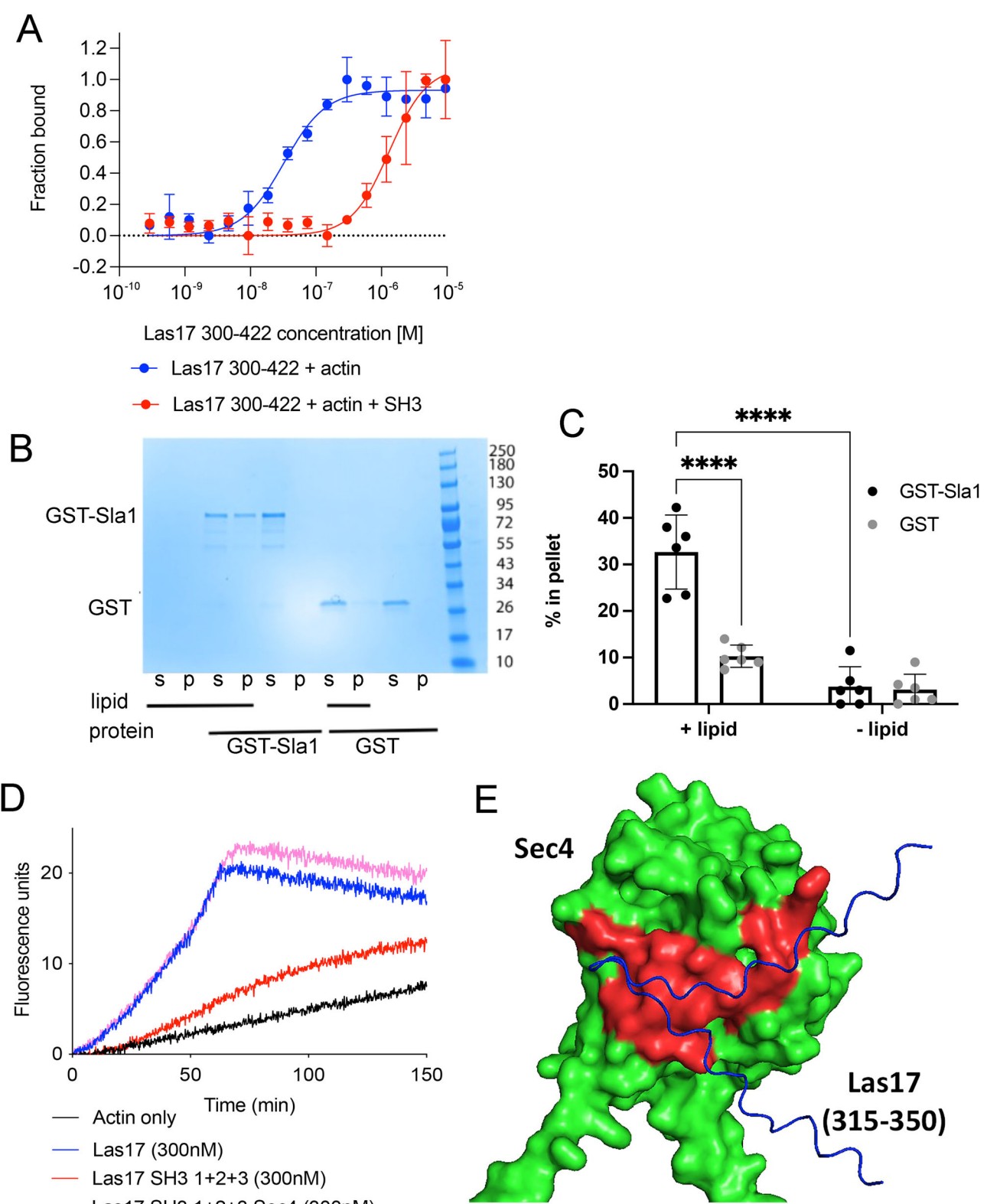

**Fig. 7 | Factors affecting Las17 interactions. A** Microscale thermophoresis showing binding of Las17 to actin in the presence (red) or absence (blue) of Sla1 SH3 domains #1–3. The concentration of actin (50 nM) and Sla1 (4.5 μM) were kept constant throughout the experiment, and the concentration of Las17 300–422 was varied between 0.29 and 9.5 μM. Error bars are standard deviation. **B** Liposome co-sedimentation assay shows that GST-Sla1 SH3#1-3 co-precipitates with liposomes prepared from bovine brain extract, whereas GST alone does not. **C** Quantification of liposome co-sedimentation assays. Šídák's multiple comparisons test P < 0.0001 ($n = 6$). **D** Representative pyrene actin assay showing alleviation of Sla1 inhibition (red) by Sec4 (pink). Actin only (black); Actin + Las17 (blue). **E** Alphafold prediction of an interaction of Sec4 with a region of Las17 primarily between 327 and 333 which lies at the C-terminal end of the first Las17 PP motif. Shown is surface representation of Sec4 in green and ribbon depiction of Las17 in blue with predicted interacting surface residues within 3.5 Å in red.

**Fig. 8 | Las17 P387 is not required for Las17 binding to Sla1 SH3 domains. A** Purified individual GST-tagged Sla1 SH3 domains were used to probe a Celluspot array consisting of a series of 12 amino acid peptides starting with residue 181, and subsequent peptides starting at two amino acid intervals, up to amino acid 540. Binding of GST-Sla1 SH3 to the array was identified by further probing of HRP-tagged anti-GST and subsequent visualisation using chemiluminescence. Peptide spots corresponding to actin binding sites on Las17 (pink), additional spots including P387A (red) and P388A (green) and their corresponding wild type peptide (blue) are also shown. Binding to the peptide containing P387A is strongly reduced or abrogated for each Sla1 SH3 domain. **B** Microscale Thermophoresis of Las17 300–422 wild type (blue), P387A (red) and P388A (green) showing that neither of these mutations affects Las17 binding to actin. Error bars are standard error of mean.

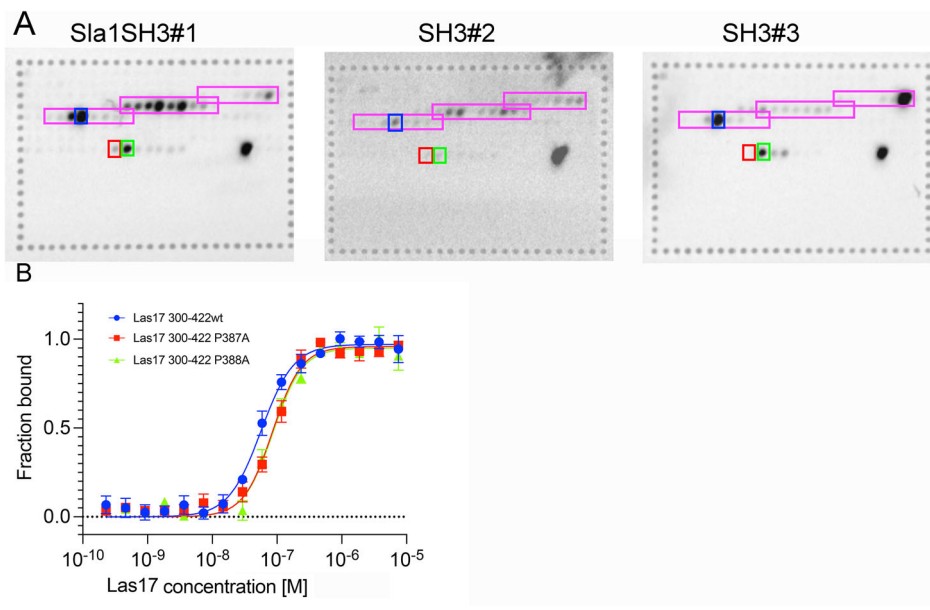

endocytosis in budding yeast in which the WASP homologue Las17 plays a central role in facilitating formation of actin filaments to support plasma membrane invagination. Understanding how actin filaments are polymerised, especially de novo, is relevant to a multitude of situations beyond yeast including at the autophagosome and on endosomes in mammalian cells[44,45].

We show that Las17 N-terminal PPR has three actin monomer binding sites and structural modeling predicts the proline tract and the flanking arginines of Las17 interact with the barbed end of G-actin. The importance of arginine residues for actin binding by Las17 has been demonstrated in vitro[15,16] while the importance of the predicted interacting glutamates in actin has been demonstrated in studies in vitro and in vivo. Mutagenised actin E334 in *act1-103* (E334A, R335A, K336A)[46] causes recessive lethality in vivo, and purified Act1-E334 has altered actin polymerization dynamics[47]. Mutant *act1-101* (D363A, E364) causes recessive temperature sensitivity and increased sensitivity to latrunculin-A in vivo[43,46]. While hydrophobic residues in actin have not been systematically mutated, deletion of the C-terminal phenylalanine (F375-ter) causes a temperature sensitive phenotype possibly indicating the importance of the extreme C-terminus in these interactions[48].

Identification of the actin binding sites with intervening spacer regions allows a model to be generated in which three monomers can bind longitudinally giving an arrangement and spacing of binding similar to that proposed for actin nucleators such as Spire[2]. Generation of an actin nucleus would require addition of a cross-linked monomer to initiate a second protofilament as proposed for Cobl[2,36]. In the case of Las17 it is possible that this could be delivered by its downstream WH2 domain and there is evidence in vitro for this contribution[15]. Taken together, data here and in previous studies demonstrate that Las17 has the capacity to generate filaments. In vivo these could serve to facilitate the activity of the Arp2/3-complex in generating the huge increase in filaments required for endocytic membrane invagination.

Having extended ideas for how Las17 could function in actin polymerisation, it is critical to understand how this can be regulated to ensure actin filaments are only generated at appropriate sites in cells. Using a more physiological ratio of Sla1-SH3 to Las17, inhibition of actin polymerisation was only observed when domains were in tandem. This was also reflected in measured binding affinities with tandem Sla1 SH3#1-2 domains having a binding affinity for Las17 PPR nearly 100-fold greater than a single SH3 domain. During yeast endocytosis many proteins recruited after initiation of

actin polymerisation have single SH3 domains which, as they accumulate, could potentially function to switch off or restrict Las17-mediated actin polymerisation activity.

Given the binding affinity of Sla1 SH3 domains for Las17, it is likely that in the cytosol, Sla1-binding completely inhibits Las17 actin polymerising function. Sla1 and Las17 are recruited to endocytic sites more than 10 seconds before Arp2/3, leading to our proposal that 'fuzzy' binding by multiple factors already at these sites releases the inhibition and facilitates Las17-mediated polymerisation of mother filaments which then recruit Arp2/3[27]. In Fig. 7 we demonstrate competitive binding of SH3 and actin for the proline tracts in Las17. We also show that the small GTPase Sec4 can reduce inhibition of Las17 by Sla1 and facilitate actin polymerisation similar to a previous study[24]. An Alphafold predicted structure (Fig. 7C) suggests that Sec4 could make extensive binding interactions with a region of Las17 including, and just downstream of, its PP1 region. This binding would lead to a weakening of Sla1 inhibition. Binding to Sec4 could then pave the way for other interactions that contribute both to release of Sla1 binding and facilitation of actin binding. Myriad binding interactions could contribute to reduced Sla1 binding of Las17. Sla2/Hip1R is an endocytic protein recruited just prior to Sla1/Las17 and binds Sla1 between its second and third SH3 domain[49]. It can also interact directly with the plasma membrane and with actin filaments[50,51]. The third SH3 domain of Sla1 can bind ubiquitin, a peptide tag added to proteins marked for endocytosis. While binding to ubiquitin is relatively weak (39 μM), a high local concentration of ubiquitin on cargoes could facilitate its binding at the membrane[10]. Downstream of Sla1-SH3#3 are binding sites for cargoes carrying NPFXD motifs and for clathrin[23,52–54]. We also identified a PH-like lipid binding site in Sla1 between SH3#2 and SH3#3 (Fig. 7B, C; Supplementary Fig. 5) and previously revealed a lipid-binding region at the N-terminus of Las17 (residues 1-180)[15]. The combined evidence from these studies leads us to propose that an initial interaction of Sla1 with Sec4 or other cargoes localized at the plasma membrane leads to its SH3 domain interactions with Las17 being effectively 'unzipped'. Unzipping can only occur when there are sufficient interactions along the length of Sla1 so effectively prevents inappropriate Las17 activity. Sla1 would concomitantly 'zip' into a different set of interactions closely associated with cargo. This switch in interactions can be seen microscopically when Sla1 along with fluorescent reporters moves inwards during endocytosis, with Sla1 decorating the tip of the invagination[55]. As Sla1 becomes unzipped, sites on the Las17 PPR become available for binding other proteins and unlike Sla1, Las17 remains in the plane of the membrane.

**Fig. 9 | The effect of *las17 P387A* mutation on endocytic reporter protein behavior. A–C** Lifetime of patches from wild type (red) or *las17 P387A* (blue) expressing cells tagged with fluorescent markers, alongside representative kymographs. **A** Las17-GFP, (**B**) GFP-Abp1, (**C**) Arc-15 mCherry (describing Arp2/3 complex lifetime). Statistical test Unpaired Mann Whitney test, **** indicates *p* value < 0.0001. A size bar for each set of kymographs is shown on the upper right of each set. For (**A**) and (**C**) size bar is 100 nm and for B is 200 nm. Error bars are standard deviation. **D** Time lapse images showing co-localisation of Las17-GFP (green dots) and Arc15-mCherry (red dots) in wild type cells and *las17 P387A* cells. The initial *las17 P387A* images contain two adjacent endocytic sites. The site of interest is labelled in the first two time panels with a white arrow. Exposure 0.5 sec with 1 second time lapse. 120 s recorded. **E** Quantification of lifetime of Las17-GFP alone (green), Arc15-mCherry alone (red) and overlap between Las17-GFP and Arc15-mCherry (yellow) from multiple endocytic patches is shown (*n* = 13). Error bars are standard deviation.

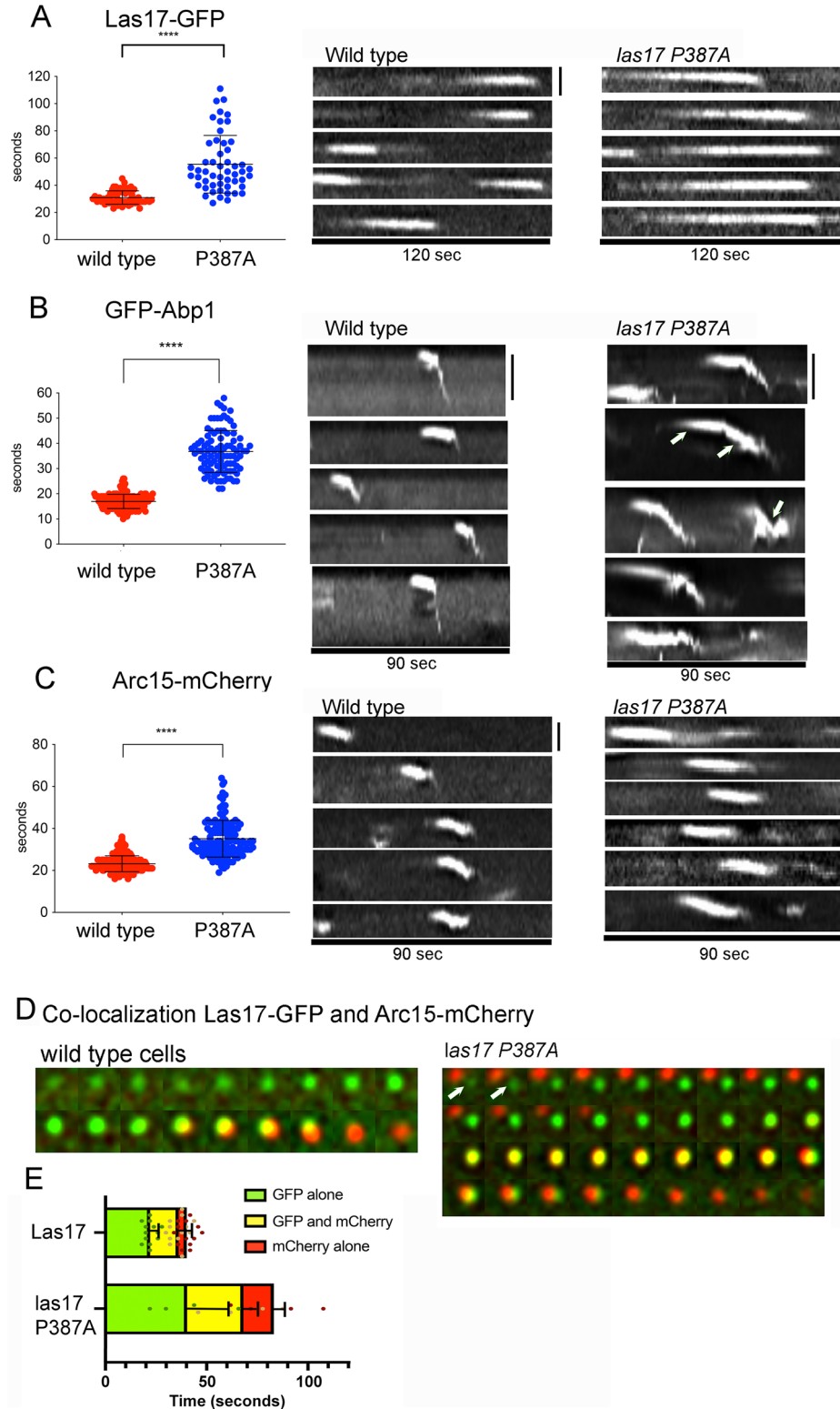

At the earliest stage of endocytosis there are no other SH3 domain proteins at endocytic sites and this would potentially allow binding of actin monomers. Successive actin binding events are likely to then shift equilibria towards formation of the actin seed required for filament formation and growth. The Las17 C-terminal PPR (PP-C; Fig. 3A) has not been shown to bind SH3 domains or monomeric actin but is able to bind F-actin[15,17]. This binding could stabilise nascent filaments and tether them near the membrane prior to them binding other F-actin binding proteins. These actin filaments would also bind and facilitate activation of Arp2/3 and subsequent stages of endocytic invagination. The generation of a las17 P387A mutant defective in binding SH3 domains at one Las17 PP motif led to aberrant actin dynamics and endocytosis further indicating the importance of balancing SH3 and actin binding in the PPR.

Understanding how actin filaments can be generated de novo and how this activity can be regulated has significant implications beyond yeast endocytosis. The demonstration that monomeric G-actin can bind in PPR

regions overlapping with SH3 domain binding gives a different way to think about regulatory functions of SH3 domains. Our data and model also highlight the likely importance of myriad weak interactions to ensure appropriate spatial and temporal regulation of key cell processes.

# Methods

## Materials
The majority of chemicals were purchased from Sigma Aldrich except where otherwise stated. Bacterial media and yeast media were from Formedium. For NMR $^{15}$N ammonium chloride and $^{13}$C labelled glucose were purchased from Cambridge Isotope Laboratories Inc (MA, USA).

## Molecular Biology
Cloning for DNA plasmid constructs used restriction enzymes and DNA ligase from New England Biolabs. Fragments were cloned using BamH1 and Sal1 enzymes except for 6×His-Las17(300-422) which used BamH1-HindIII. Oligonucleotides were purchased from Operon Technologies Inc. Generation of various mutants was achieved using the QuikChange mutagenesis kit (Stratagene) according to manufacturer's instructions. Plasmids used are listed in Supplementary Table 1. All new constructs were verified by sequencing using University of Dundee sequencing facility (https://dnaseq.co.uk/).

## Protein purification
Rabbit skeletal muscle actin was gel filtered as described[56]. Concentrations of G-actin produced and used in experiments were in the range 35–65 μM.

Las17 protein fragments were expressed in *E. coli* and purified as described[15]. Briefly, GST-expression plasmids were transformed into Lucigen OverExpress C41(DE3) cells. Protein was purified from 2 litres of cells after induction by isopropyl β-D-1-thiogalactopyranoside (5 hr 37 °C unless stated otherwise). Lysates were incubated with glutathione Sepharose beads (GE Healthcare) for 1 h at 4 °C, washed, and proteins were cleaved using PreScission Protease (Cytiva, Little Chalfont, UK) for no more than 6 h at 4 °C. Cleaved protein was buffer exchanged into G-buffer (2 mM Tris-HCl [pH 8.0], 0.2 mM CaCl$_2$, 1 mM NaN$_3$, 0.5 mM dithiothreitol, 0.2 mM ATP), kept on ice at 4 °C, and used within 24 hr. Las17 (342–392) was purified using a GST tag, and left on the tag as it was so small. In Fig. 4A there was a small effect of GST only on the actin polymerisation rate. This small increase in rate was subtracted from the Las17 342–392 rate to simplify the figure. Raw data prior to this subtraction is provided.

Sla1-SH3 protein domains were expressed in *E. coli* and purified as described for Las17 protein domains, except that for single domains the GST tag was removed using an overnight incubation with thrombin at room temperature, in thrombin cleavage buffer (20 mM Tris pH 8.4, 150 mM NaCl, 2.5 mM CaCl$_2$). Tandem SH3 domain fragments were cleaved with Prescission protease as above.

His-tagged Sla1-SH3 domains and Las17 fragments were expressed in *E. coli* Rosetta (DE3) Competent Cells. Two litres of cell culture were induced with isopropyl β-D-1-thiogalactopyranoside (18 hr 17 °C). After cell breakage, lysates were pushed through a 1 mL nickel column (Cytiva HisTrap HP Prepacked Columns, Fisher) and washed with increasing levels, up to 100 mM imidazole. Protein was eluted using 500 mM imidazole and exchanged into the appropriate buffer. This included either G-buffer (2 mM Tris-HCl [pH 8.0], 0.2 mM CaCl$_2$, 1 mM NaN$_3$, 0.5 mM dithiothreitol, 0.2 mM ATP) or PBS (137 mM NaCl, 10 mM Na$_2$HPO$_4$ [pH 7.4], 2.7 mM KCl, 1.8 mM KH$_2$PO$_4$).

## Protein assays
*Fluorometry assays* were performed as described previously[15]. 370 μl assays used 3 μM actin as indicated unless otherwise specified. Pyrene-actin was added to 5%, mixed thoroughly, and added to the fluorometry cuvette. Polymerization salts, 10× KME (500 mM KCl, 10 mM MgCl$_2$, 10 mM EGTA, 100 mM Tris-HCl [pH 8.0]) were mixed with fragments or G buffer to give final concentration of 0.5× KME before reading fluorescence. Polymerization was observed in a Cary Eclipse fluorometer (excitation 364 nm, slit 10 nm

round; emission 385 nm, slit 20 nm) as described previously[15,17]. Unless otherwise stated Las17 fragments were used at 0.3 μM and Arp2/3 (Cytoskeleton Inc, USA) at 5 nM. Experiments were carried out three separate times with different protein preparations and a representative graph shown.

*Microscale Thermophoresis* (MST) assays were performed as described previously[15]. Briefly, rabbit muscle actin in G-buffer minus DTT was lysine labelled using Monolith NT$^{TM}$ Protein Labelling Kit RED-Malemide (NanoTemper Technologies, Munich, Germany). Las17 was prepared as a two-fold serial dilution across 16 tubes and an equal volume of labelled actin was added to each reaction (Final buffer conditions: 25 nM actin, G-buffer + 0.05% Tween). Each reaction was then loaded into Monolith NT.115 Premium Treated Capillaries (NanoTemper Technologies) and thermophoresis was measured in a Monolith NT.115 Microscale Thermophoresis device (NanoTemper Technologies). Thermophoresis was measured at 20% red LED power and 40% IR laser power at 22°C. Fnorm was calculated from the ratio of fluorescence before heating and after the new equilibrium had been reached. Fnorm was then plotted against Las17 concentration to give the Las17 dependent change in thermophoresis. This was described by the 'K$_d$ fit' model on the provided software to give K$_d$s for actin binding to Las17 fragments. Each experiment was carried out three separate times with protein from different protein preparations and mean ± SE shown except for Fig. 7a where mean ± SD is shown.

*Biolayer Interferometry* (BLI) assays were performed using an Octet® R2 (Sartorius). Protein samples were placed into the wells of a polystyrene plate (ProxiPlateTM, Perkin Elmer) and binding measured either with Octet NTA Biosensors or Octet GST Biosensors. Screening assays were performed to determine the optimum loading concentration and time for each binding pair to ensure that the probe was not saturated. All experiments were repeated three times, each time with newly prepared protein. Analysis and visualisation were performed using GraphPad Prism 10.0.2.

*Liposome binding assays* were carried out as described previously[15]. To prepare liposomes 11 μl of a 25 mg/ml solution of Folch fraction-1 was dried under a nitrogen stream. They were washed in chloroform, dried under nitrogen, and further washed with diethyl ether to remove all traces of chloroform before being dried again. The dried lipid mixture was resuspended in 200 μl of buffer B (20 mM HEPES pH 7.2, 100 mM KCl, 2 mM MgCl$_2$, 1 mM DTT) at 37 °C for 30 min with regular gentle agitation. Liposomes were then extruded 11 times through polycarbonate filters with 1.0 μm pores. For liposome binding assays 20 μM Sla1 (pre-spun at 313,000 × *g* for 15 min) was mixed with 20 μl liposomes in buffer B, (final volume 50 μl) and incubated at room temperature for 15 min. Liposomes and bound protein were pelleted by centrifugation at 250,000 × *g* for 15 min, and supernatants and pellets carefully separated. The pellets were resuspended in 50 μl buffer B and samples were analysed by SDS-PAGE. The proportion of protein in each sample was determined by densitometry of the Coomassie stained gels using a BioRad GelDoc. Replicate experiments were carried out with protein from different preparations. To normalise the data the amount of protein in the supernatant and pellet of each sample was added together to give a value for 100%, and then the % in either the supernatant or pellet alone was calculated. When liposomes were included in the pyrene actin assay they were prepared as described for liposome binding assays, but the dried lipid mixture was resuspended in 5×KME in place of liposome buffer. This was then used as a source of KME in the pyrene actin assay as well as a source of lipid. Mean ± SD is shown n = 6. Sîdák's multiple comparisons test was carried out using Graphpad Prism software, **** denotes two tailed P value of < 0.0001.

*Membrane dot blots* carrying overlapping 12-mer peptides of Las17 over the region 300-536 were purchased from Intavis Celluspots. They were incubated with GST fused SH3 domains of Sla1 or with GST alone, as described in ref.[57]. Membranes were probed with HRP tagged anti-GST, and the presence of SH3-GST fusion protein was detected with ECL chemiluminescent detection solution (Thermo Fisher Scientific).

*Actin filament growth rate assays.* Total Internal Reflection microscopy (TIRFm) was performed using a method modified from[58] using purified rabbit muscle actin and Alexa488 actin (Cytoskeleton, Inc) at a 4:1 ratio.

Glass coverslips (Deckgläser) were cleaned using a modification of the method described[59]. Briefly, coverslips were rinsed with gentle agitation in acetone (30 min), then water (5 min) then cleaned in Piranha solution (13 ml concentrated sulphuric acid and 6 ml hydrogen peroxide) for 90 min before rinsing with distilled water and ethanol, drying and coating with 1 mg/ml mPEG silane MW 2000 (Layssan Bio). TIRF reaction buffers 10× KMEI, GOC mixture, and methyl cellulose (400 cP, Sigma) were prepared in advance. 10× KMEI buffer (500 mM KCl, 100 mM Imidazole (pH 7.0), 10 mM MgCl$_2$, 10 mM EGTA (pH 7.0)) was filtered through a sterile syringe-filter device (0.22 μm, Thermo Fisher Scientific) and stored at room temperature. GOC mixture was prepared by dissolving 20 mg of Catalase (Calbiochem) and 100 mg of Glucose-oxidase in 10 ml cold water. Undissolved particles were cleared by ultracentrifugation and 50 μl aliquots from the upper 90% of the supernatant were stored at −20 °C. 50 ml of 2% methyl cellulose was prepared by dissolving 1 g methylcellulose in 25 ml of Milli-Q water preheated to 65 °C, before adding 25 ml of room temperature water and rotating overnight at 4 °C. This was ultracentrifuged (150,000 × g) and upper 75% of the supernatant transferred to a fresh tube and stored at 4 °C.

Immediately before use 2× TIRF buffer was prepared by mixing 1 volume of 1× KMEI with 0.01 volumes of GOC mixture and 0.01 volumes of 2% methylcellulose. A prepared TIRF chamber was pre-incubated with 1× TIRF buffer containing 5% BSA. The TIRFm reaction was set up in two separate tubes each containing a subset of the total reaction constituents at 2× final required concentration. Actin in 2x G-buffer in one tube, and the second containing Las17 (and Arp2/3 if required) in 2× TIRF buffer. The reaction was initiated by mixing 20 μl of each of these solutions to give final assay conditions of 0.5 μM actin and 10 mM imidazole pH7.4, 25 mM KCl, 0.5 mM MgCl$_2$, 0.5 mM EGTA, 0.2 mM CaCl$_2$, 0.2 mM DTT, 0.1 mM ATP, 15 mM glucose, 0.0002% methylcellulose, 10 μg/ml catalase, 50 μg/ml glucose-oxidase. The time was noted and the sample was added to one side of the TIRF chamber and allowed to run under the coverslip and into the chamber. Actin polymerisation was observed on a Nikon eclipse Ti inverted microscope, set up for TIRF illumination, at 3% laser power and 30 ms exposure time. Images were captured on an iXon ultra EMCCD camera (Andor). Experiments were carried out four times with protein from different preparations. Mean ± SD is shown, number of replicate experiments is 4. Total number of individual samples: actin only n = 35, actin + Las17 $n = 37$, Actin + Las17 + Arp2/3 $n = 43$. Ordinary one-way anova statistical analysis was carried out using Graphpad Prism software, **** denotes one tailed P value of < 0.0001.

## Structural modelling

Modelling of binding between Las17 and actin was performed with a PDB file containing yeast *ACT1* actin (PDB: 1YAG) and the Las17 peptide covering ABS3 (PQATG**RRGPAPPPPP**RASRP) using HPEPDOCK2[33] or FlexPepDock[60,61]. HPEPDOCK 2.0 modelling was submitted 20/09/2019 whilst FlexPepDock modelling was submitted 31/05/2021. The top 10 predicted structures from each were noted. HPEPDOCK was shown to perform best in a comprehensive review of 14 docking programs[32]. PIPER-FlexPepDock uses rigid-body docking followed by flexible refinement of the peptide, HPEPDOCK first addresses the peptide flexibility by generating an ensemble of configurations before rigid docking each in turn[33,34].

*Alphafold analysis.* Sla1 (Uniprot P32790) was viewed in Alphafold2.0 https://Alphafold.ebi.ac.uk/[28] and folded domains compared to known domains using the *Saccharomyces* genome database (https://www.yeastgenome.org/). Colabfold analysis between Sec4 and Las17 used the structure 1G17.pdb with ligand comprising residues 300–422 of Las17[62] (https://colab.research.google.com/github/sokrypton/ColabFold/blob/main/Alphafold2.ipynb). Pymol was used to visualise and analyse domains and possible interactions further (https://www.pymol.org/).

## NMR experiments

Experiments were carried out on Bruker DRX-600 and DRX-800 spectrometers fitted with cryoprobes. SH3 domains 1 and 2 were assigned using a standard set of HNCA/HN(CO)CA, HNCO/HN(CA)CO and CBCAHN/CBCA(CO)HN 3D spectra. The assignment has been deposited at BioMagResBank, with code 52650. SH3 domain-3 was assigned by comparison to data provided by Prof Ishwar Radhakrishnan (Northwestern University) for the same domain although with different residues at both ends. Titrations were conducted with three peptides (GenScript Biotech, UK) comprising the three PP motifs (Fig. 3A): p1 RNNRPVPPPPPP MRTTTEGSGVR (residues 318–339); p2 RTTTEGSGVRLPAPPPPP RRGPAPPPPPPHRHV (residues 330–361); and p3 LPAQTGRRG-PAPPPPPPRASRPTP (residues 375–397). Peptide solutions were prepared as concentrated stock solutions in the same buffer at the same pH and added stepwise to protein solutions. Shift changes were analysed by picking peaks in Felix (Felix NMR Inc, San Diego, CA), which were downloaded to text files and for determining $K_d$ were analysed using Levenberg-Marquardt least squares fitting to the standard quadratic equation[63]. Analysis of the $^{15}$N relaxation data was carried using CCPN Analysis[64].

## Yeast strains, growth and staining

Yeast strains used in this study are listed in Supplementary table 2 and plasmids used in Supplementary table 1. Unless stated otherwise, cells were grown with rotary shaking at 30 °C in liquid YPD medium (1% yeast extract, 2% Bacto-peptone, 2% glucose supplemented with 40 μg/ml adenine) or in synthetic medium (0.67% yeast nitrogen base, 2% glucose) with appropriate supplements. All strains carrying fluorescent tags have growth properties similar to isogenic control strains with the exception of strains carrying dual reporters where lifetimes were longer for both wild type and mutant strains (in Fig. 9D). Point mutations in *LAS17* were generated using site directed mutagenesis (Agilent). DNA cassettes carrying mutations for integration were transformed into KAY1801. Mutant colonies were counter-selected on minimal medium, containing 0.005% uracil and 0.1% 5'-Fluoroorotic Acid (5-FOA; Melford laboratories). Allele exchange, in growing Ura3⁻, 5-FOA resistant colonies, was confirmed by PCR and sequencing.

For live-cell imaging, cells expressing tagged proteins were visualised after growing to early log phase. Analysis was carried out at room temperature (22 °C). Time-lapse of wild-type and mutated Las17-GFP was acquired using OMX DeltaVision V4 and a 60×/1.42 NA objective. Images were taken simultaneously on separate scientific complementary metal oxide semiconductor (sCMOS) cameras (30 msec exposure). Seven 250 nm sections were acquired every 0.5 sec (241 time points). The stacks were then deconvolved and processed, using SoftWorx. Protein localisation and lifetime was analysed from those projections. Fiji software was used to assemble movies and kymographs, to track and record behaviour of individual spots. Graphs show mean ± SD. Sample size Fig. 9A wt $n = 67$, P387A $n = 52$; 9B wt $n = 183$, P387A $n = 96$; 9 C wt $n = 156$, P387A $n = 132$. Where appropriate data was tested for normal distribution using Graphpad Prism software, and then subjected to the appropriate test as described in the methods for each experiment type.

## Statistics and reproducibility

*MST assays:* Each experiment was carried out three separate times with protein from different protein preparations and mean ± SE shown except for Fig. 7a where, for clarity, mean ± SD is shown. *Protein fluorometry assays* and *Biolayer Interferometry (BLI) assays* were carried out three separate times each with different protein preparations and a representative graph shown.

*Liposome binding studies:* triplicate experiments were carried out with protein from different preparations. To normalise the data the amount of protein in the supernatant and pellet of each sample was added together to give a value for 100%, and then the % in either the supernatant or pellet alone was calculated. Mean ± SD is shown n = 6. Sîdák's multiple comparisons test was carried out using Graphpad Prism software, **** denotes two tailed P value of < 0.0001.

*TIRF microscopy assays:* Experiments were carried out four times with protein from different preparations. Mean ± SD is shown, number of replicate experiments is 4. Total number of individual samples: actin only $n = 35$, actin + Las17 $n = 37$, Actin + Las17 + Arp2/3 $n = 43$. Ordinary one-way anova statistical analysis was carried out using Graphpad Prism software, **** denotes one tailed P value of < 0.0001.

*Lifetime and behaviour of cell reporter proteins using fluorescence microscopy*: Graphs show mean ± SD. Sample size Fig. 9A wt $n = 67$, P387A $n = 52$; 9B wt $n = 183$, P387A $n = 96$; 9 C wt $n = 156$, P387A $n = 132$. Unpaired Mann Whitney statistical analysis of patch lifetimes was carried out using Graphpad Prism software, **** denotes two tailed P value of <0.0001.

## Reporting summary

Further information on research design is available in the Nature Portfolio Reporting Summary linked to this article.

## Data availability

Data supporting the findings of this work are available within the paper and in the Supplementary Information files e.g. Supplementary Fig. 7 is the full gel file for the gel in Fig. 8. Numerical data files for the study are available as Supplementary data files: Supplementary data 1 has data for Fig. 1; Supplementary data 2 has data for Fig. 2b; Supplementary data 3 has data for Figs. 2c and 3d; Supplementary data 4 has data for Fig. 3d; Supplementary data 5 has data for Figs. 4, 6, 7, 8, 9 and Supplementary data 6 has information about the peptide spot array used in Fig. 8. Materials from this study are available on request without restriction. New plasmids generated in this study are deposited with Addgene: pKA1284 – ID239367; pKA1247 – ID239368; pKA1248 – ID239369; pKA1280 – ID239370; pKA1317 – ID239371; pKA1336 – ID239372; pKA1252 – ID239373; pKA1278 – ID239374; pKA1279 – ID239375; pKA1337 – ID239376.

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

## Acknowledgements

We are grateful to Professor Jason King and Dr Joe Tyler for critical reading of the manuscript. We are deeply grateful to Prof Ishwar Radhakrishnan (Northwestern University) for providing assignments for a closely related protein and to Professor Jon Sayers for use of the BLI. K.R.A discloses support for this work from BBRSC [BB/N007581/1] and The Leverhulme Trust [2021-099\2]. M.P.W. and K.R.A. disclose support from BSBRC [BB/M011151/1]. M.P.W. discloses support from BSBRC BB/R000727/1.

## Author contributions

K.R.A., E.G.A. and J.S.P. conceptualised the study. K.R.A., E.G.A., J.S.P., L.P.H., I.I.S. and M.P.W. were involved in experimental design. L.P.H., J.S.P., E.G.A. and A.J.H. optimised methodology and purified all proteins used in the study. L.P.H, J.S.P and E.G.A. performed the pyrene and MST assays; L.P.H. undertook Bilayer interferometry; J.S.P. undertook TIRF assays; E.G.A. performed SPOTS assays and liposome binding; M.L.R. and M.P.W. performed all nmr experiments and analysis; L.P.H. and K.R.A. undertook structural modelling; I.I.S. generated strains and performed the experiments in yeast. K.R.A. wrote the manuscript. All authors contributed to reviewing and editing of the manuscript. L.P.H. and J.S.P. contributed equally to the manuscript. K.R.A. is the corresponding author.

## Competing interests

The authors declare no competing interests.
