## [Transparent Peer Review file · Communications Biology]

Competitive Binding of Actin and SH3 domains at Proline-rich regions of Las17/WASP Regulates Actin Polymerisation

Corresponding Author: Professor Kathryn Ayscough

Version 0:

Reviewer comments:

Reviewer #1

(Remarks to the Author)

The authors discovered that Nter PPR can bind multiple actin monomers and promote actin polymerization. This process is regulated by the binding of the Sla1 SH3 domain, particularly the tandem Sla1 SH3 domains. I appreciate the significance of this work, which may interest a broad community studying the actin cytoskeleton in yeast. However, I believe the authors should make an effort to help readers better understand their data and concepts.

Major points:

1. The authors experimentally demonstrated that three sites in the Las17 PPR contribute to actin binding, and their modelling also supports this hypothesis. However, the experiments were conducted with mammalian actin, while the modelling was performed using yeast actin. It is widely accepted that actin is highly conserved in eukaryotes, and rabbit muscle actin is commonly used for most in vitro research. The authors should acknowledge this and discuss the reliability of their data. For instance, they could address whether the residues at the interface predicted by the modelling are conserved between yeast and mammalian actin.

2. The structure figures have significant room for improvement. I advise the authors to read some structural biology papers or/and seek advice from structural biologists on how to properly present 3D protein structures. They can also find useful information on PyMOLwiki.

- Fig 2A. the interface hydrogen bond distance are hardly readable. Every label should be readable; otherwise, there is no point of putting a label. If the authors wish to show both the overall structure and the interface, I suggest adding a zoomed-in figure in the right panel. Additionally, I think a dark dashed line, rather than bright yellow, would be more readable against the white background.

- Fig 5A. Not readable. Is it possible to merge A and B by making a zoom in figure?

- Fig 5B. I have no clue what these yellow dash lines are. Again, labels are not readable. The authors did not give the information regarding R6 and R7.

- Fig 5C. It is possible to adjust the label placement in PyMOL to prevent it from overlapping with the residues.

- Fig 5D. Instead of manually placing these selected residues, the authors could simply connect the red residues with a dashed line. This approach is commonly used by structural biologists to indicate regions where structural information is missing. More importantly, it is misleading to position the actin in this way and to suggest that the trimeric actin arrange as actin seed (see Major point 3.)

- Fig 7E. What are those yellow dashes? This is not informative at all. Instead, the authors can show the surface of the interface. Or they can show the residues with distance between say 3 angstrom and 4 angstrom.

- There are similar issues in supplementary figures with some legends are not understandable (for example residues with shifter changes larger than ??).

3. There is no evidence showing that the N-terminal three-actin-binding PPR can arrange actin monomers into a trimeric actin seed. If it could, the authors would observe Las17 (300-392) accelerating actin polymerization in the absence of Arp2/3 in the pyrene assay. They would also observe more nucleated actin filaments (instead of or in addition to fast elongating actin filaments) in the TIRF assay.

Minor points:

1. The authors should put forward that the molecular function of the N-terminal PPR is different from that of the C-terminal PPR in the introduction instead of only talking about N-ter PPR.

2. The naming system is not consistently followed throughout the paper. For instance, mutant names should be italicized, but in lines 360–361, 368, and 396, *las17 P387A* is not italicized. In these cases, the *l* is lowercase; while in line 358, the *L* of *Las17 P387A* is capitalized and in line 331 *Sla1 P387A*. Please review the entire paper to ensure that the yeast naming conventions are consistently applied.
3. When 'actin' is mentioned in the paper, it is not always clear whether it refers to G-actin, F-actin, or both. For example, the MST data shows that *Las17* binds to G-actin, as the experiments were conducted in G-buffer and with low actin concentrations. However, I believe the authors should explicitly state that the experiments were performed with G-actin to ensure that readers without expertise in actin biochemistry can easily understand this point.
4. In Fig 1A, the red PP representing Poly Proline is misleading. It looks like two Proline aa. Probably put 'PP' in rectangles to represent motifs?
5. Fig 3A should definitely go before Fig1A, at least the diagram of the *Las17* domains. Fig 3A contains plenty of information. But the way it presents is very confusing. Why PP motif 1 is coloured in pink and the other two PP motifs are coloured in red? Why 342 and 392 were highlighted in this way? Should PP-N be PPR-N, or N-PPR referring to N-terminal polyproline region?
6. Is there a specific reason to use 0.5 x FME instead of 1 x FME for the pyrene assay? Also, please double check the emission and excitation wavelength of pyrene assay.

Reviewer #2

(Remarks to the Author)

[Summary]

Hancock et al. investigated the regulation of *Las17* by *Sla1* and *Sec4*. Using NMR, pyrene assays, MST, and AlphaFold predictions, they identified a crucial actin-binding domain in *Las17* that overlaps with the *Sla1*-SH3 domains. This finding explains how *Sla1*-SH3 inhibits *Las17*-induced polymerization promotion.

The data presented in the manuscript are of good quality and support the authors' findings. While I recommend the study for publication in *Communications Biology*, the logical progression from data to conclusions is not clearly explained, making the argument difficult to follow. This may be due to some conclusions within paragraphs incorporating findings from later figures. To improve clarity, the authors should carefully explain which aspects of each figure support specific conclusions

[Major Comments]

1. Line 173-178: The authors should describe Figures 2B and 2C more carefully to guide readers in understanding, including how the local mobility increase of D52-E55 and E67-V70 have same correlation time are derived.
2. Line 185-189: The authors should elaborate on the results related to Figures 3B, 3C, and 3D to guide readers in understanding, including why the authors concluded all three PP peptides of *Las17* bind to SH3#1 and SH3#2 of *Sla1*. Providing the K_d values would also be helpful.
- 3-1. Line 193-196: The description explaining why the 28 aa spacing is the optimal choice, with SH3#1 binding either PP2 or PP3 and SH3#3 binding PP1, is missing.
- 3-2. Also, why does the affinity of SH3#1&2 become 100 times tighter than SH3#1 alone (Fig. 1C), even though both SH3#1 and SH3#2 bind to PP2 or PP3 without any preference?
4. Description of Figure 3E is missing.
- 5-1. Line 231-234: The authors have not fully addressed the point in the following sentence: 'The designation of three distinct actin binding sites (ABS1, ABS2, and ABS3) lying adjacent to the PP motifs 1, 2, and 3, respectively.'
- 5-2. Additionally, the PP motif (the *Sla1* binding site) should be included in ABS1, rather than being described as adjacent?
6. Does the model in Figure 5D meet the binding affinity estimated in Figure 4B?
- 7-1. Line 289: Please write down the binding affinities of both reactions, so 40-fold increase is more supported.
- 7-2. Also, what level of competition do the authors anticipate? The authors mention that the ternary complex of *Las17*: actin: SH3 may form. To support this hypothesis, even when SH3#3 dissociates, the affinity remains at 87 nM (Fig. 1C), which may still be sufficient to form the ternary complex. I believe that a careful explanation of Figure 7A will help clarify the authors' reasoning. The authors may be able to use the data on *Sec4*-induced inhibition of SH3 binding to *Las17* (Fig. 7D) to help explain why ABS1 is the only essential domain for *Las17* in the solution assay.
8. What is actin concentration in Figure 7D? Is the assay identical to the other pyrene assays in this manuscript?
9. How was the lifetime quantified in Figures 9A, 9B, and 9C? The lifetime of *Las17 P387A* appears shorter compared to the kymograph in the left panel. It may be helpful to add lines to the kymographs to indicate how the lifetime was determined. Additionally, please include the Y-axis scale in the kymograph.

[Minor concerns]

Line 92: 'it' appears to be a typo?

Line 173: 'localized' to be a typo?

Figure S4B legend: 'Actin; 300 uM' must be a typo.

Line 433: The phrase 'generate filaments' is not precise. Consider using 'promote actin polymerization' instead.

In the Discussion, I respect the authors' passion for their consideration of a set of regulatory streams involving multiple factors. In current manuscript, however, I would suggest focusing solely on the protein-protein interactions and regulatory functions in the molecular level based on the experimental results presented in this manuscript.

Version 1:

Reviewer comments:

Reviewer #1

(Remarks to the Author)

There are two major points remain unaddressed:

- I am not convinced (yet) that full length Las can nucleate actin filaments by itself.

In the Allwood et al. paper that the authors mentioned, they found 300-422 and 300-536 helps nucleation (Allwood et al. Fig 6C) but it seems the longer the peptides are the capacity of nucleation decrease and capacity of elongation increase. It is still unclear that if the non-truncated Las can help nucleation.

The other issue is that the pyrene assay can only measure actin polymerization and cannot distinguish between nucleation and elongation of actin filaments. The authors claim that Las can induce a signal increase in the pyrene assay. They also demonstrated that Las can enhance the elongation rate in the TIRF assay. Without clear data ruling out the possibility that the pyrene signal increase is solely due to accelerated actin elongation, I remain unconvinced that full length Las17 can facilitate actin nucleation.

The authors claim that the TIRF assay cannot be used to observe actin nucleation because short actin filaments cannot be distinguished from a seed or an actin monomer. However, they were performing live imaging, and the filaments would grow, right? Eventually, by examining images taken later, they should be able to determine whether the initial dot was a real filament. In fact, the TIRF assay has been widely used to study the nucleation of actin-related proteins (Graziano et al., MBoC 2011; Sun et al., Nat Comm 2021; Mu et al., NCB 2019 ...).

-Fig 5D. This does not constitute scientific results. If the authors want to demonstrate that there is enough space between two actin monomers for the linker, a more scientific approach would be to: (1) measure and provide the distance between two monomers in an actin seed or filament, (2) determine the number of residues present in the linker, (3) assume that the linker is completely flexible, and (4) compare the length of an xx-long peptide with the measured distance to conclude whether there is sufficient space for the linker to fit between the two actin monomers. Even the flexible loop does not fold arbitrarily. There are rules to follow to avoid clashes between residue side chains, to respect backbone dihedral space etc. Therefore, I cannot accept manually placed peptides as a part of main results.

Reviewer #2

(Remarks to the Author)

Although some responses still do not fully address my comments, the content is now clearer. I recommend that this study is worthy of publication.

-Line 1142-1143: Add explanation about black and blue lines.

-Line 1178: 200 nM should be 200 nm (lower case).

-Line 1182-1183: Las17 and Arc15 should be Las17-GFP and Arc15-mCherry, respectively.

Version 2:

Reviewer comments:

Reviewer #1

(Remarks to the Author)

1. The results with TIRF assays seem convincing.

2. Regarding Fig. 5D, I believe it is standard practice in the field to use a straight dashed line to represent regions with no structural information, as I suggested in the first round of revisions. This convention is as intuitive to structural biologists as using dashed lines for bonds and solid lines for clashes.

COMMSBIO-25-0107-T

Hancock et al

Point by point response to reviewers.

For clarity, all reviewer comments are in black and our responses are in red.

We thank the reviewers for their thoughtful and insightful comments and value their input which will undoubtedly improve the manuscript and make it more widely accessible to readers.

Reviewer #1 (Remarks to the Author):

The authors discovered that Nter PPR can bind multiple actin monomers and promote actin polymerization. This process is regulated by the binding of the Sla1 SH3 domain, particularly the tandem Sla1 SH3 domains. I appreciate the significance of this work, which may interest a broad community studying the actin cytoskeleton in yeast. However, I believe the authors should make an effort to help readers better understand their data and concepts.

>We have rephrased parts of the text to improve the clarity of the overall narrative and also to support conclusions more effectively. Changes have been made throughout and are on the text document but not all included here.

Major points:

1. The authors experimentally demonstrated that three sites in the Las17 PPR contribute to actin binding, and their modelling also supports this hypothesis. However, the experiments were conducted with mammalian actin, while the modelling was performed using yeast actin. It is widely accepted that actin is highly conserved in eukaryotes, and rabbit muscle actin is commonly used for most in vitro research. The authors should acknowledge this and discuss the reliability of their data. For instance, they could address whether the residues at the interface predicted by the modelling are conserved between yeast and mammalian actin.

>We have modelled both yeast and mammalian actin with Las17 but showed here the interaction between the yeast proteins as we felt only considering binding between the mammalian actin and yeast Las17 would give a more limited perspective. In terms of the main predicted interacting residues E334, E361 and E364 all are conserved between yeast and mammalian actin (see figure below). We have previously undertaken biochemical assays that demonstrated that yeast Las17 has the same activity towards yeast and mammalian actin (e.g. Allwood et al, 2016).

Alignment **S.cerevisiae ACT1 NP_001091.1; Human ACTA KZV11559.1.**

```

NP_001091.1      MCDEDETTALVCDNGSGLVKAGFAGDDAPRAVFPSIVGRPRHQGVMVGMGQKDSYVGDEA      60
KZV11559.1      --MDSEVAALVIDNGSGMCKAGFAGDDAPRAVFPSIVGRPRHQGIMVGMGQKDSYVGDEA      58
                  :.*.:*** *****: *****:*****:*****:*****:*****:*****

NP_001091.1      QSKRGILTLKYPIEHGIITNWDDMEKIWHHTFYNELRVAPEEHPTLLTEAPLNPKANREK      120
KZV11559.1      QSKRGILTLRYPIEHGIVTNWDDMEKIWHHTFYNELRVAPEEHPVLLTEAPMNPKNREK      118
                  *****:*****:*****:*****:*****:*****:*****:*****:*****

NP_001091.1      MTQIMFETFNVPAMYVAIQAVLSLYASGRTTGIVLDSGDGVTHNVPIYEGYALPHAIMRL      180
KZV11559.1      MTQIMFETFNVPAFYVSIQAVLSLYSSGRTTGIVLDSGDGVTHVVPYAGFSLPHAILRI      178
                  *****:***:*****:*****:***** ***** *:*****:***

NP_001091.1      DLAGRDLTDYLMKILTERGYSFVTTAEREIVRDIKEKLCYVALDFENEMATAAASSSLEK      240
KZV11559.1      DLAGRDLTDYLMKILSERGYSFSTTAEREIVRDIKEKLCYVALDFEQEMQTAAQSSSIEK      238
                  *****:***** *****:*****:*****:*** ***:***:***

NP_001091.1      SYELPDGQVITIGNERFRCPETLFQPSFIGMESAGIHETTYNSIMKCDIDIRKDLYANNV      300
KZV11559.1      SYELPDGQVITIGNERFRAPEALFHPSVLGLESAGIDQTTYNSIMKCDVDVRKELYGNIV      298
                  *****:***:***:***:***:*****:*****:***:***:***

NP_001091.1      MSGGTTMYPGIADRMQKEITALAPSTMKIKI IAPPERKYSVWIGGSILASLSTFQQMWIT      360
KZV11559.1      MSGGTTMFPGIAERMQKEITALAPSSMKVKI IAPPERKYSVWIGGSILASLSTFQQMWIS      358
                  *****:***:*****:***:*****:*****:*****:*****:*****:

NP_001091.1      KQEDEAGPSIVHRKCF      377
KZV11559.1      KQEDESGPSIVHHKCF      375
                  *****:*****:***

```

2. The structure figures have significant room for improvement. I advise the authors to read some structural biology papers or/and seek advice from structural biologists on how to properly present 3D protein structures. They can also find useful information on PyMOLwiki.

- Fig 2A. the interface hydrogen bond distance are hardly readable. Every label should be readable; otherwise, there is no point of putting a label. If the authors wish to show both the overall structure and the interface, I suggest adding a zoomed-in figure in the right panel. Additionally, I think a dark dashed line, rather than bright yellow, would be more readable against the white background.

>Amended as per suggestion and new figure 2A included

Fig 5A. Not readable. Is it possible to merge A and B by making a zoom in figure?

>Figure 5A shows G-actin in the conventional orientation as we felt it was more helpful for readers to have a clear view of binding in the context of the whole actin molecule and in a way that most readers would be familiar with. We have however simplified the figure so each of the predicted peptide docking arrangements are more easily visualised. We have kept Figure 5B separate as the orientation of actin is rotated for clarity We have amended as indicated in point below

- Fig 5B. I have no clue what these yellow dash lines are. Again, labels are not readable. The authors did not give the information regarding R6 and R7.

>Amended. Specific mention is made of R6 and R7 and predicted bonds showed in black. The peptide sequence is given in the methods.

- Fig 5C. It is possible to adjust the label placement in PyMOL to prevent it from overlapping with the residues.

>Amended

- Fig 5D. Instead of manually placing these selected residues, the authors could simply connect the red residues with a dashed line. This approach is commonly used by structural biologists to indicate regions where structural information is missing. More importantly, it is misleading to position the actin in this way and to suggest that the trimeric actin arrange as actin seed (see Major point 3.)

> We feel that it is important to show the residues between the interacting proline tracts as it is only with this region present that it can be demonstrated that the tract spacing is sufficient for the binding of the three actin monomers. This is particularly relevant for researchers who are working on other actin nucleators.

New figure 5

- Fig 7E. What are those yellow dashes? This is not informative at all. Instead, the authors can show the surface of the interface. Or they can show the residues with distance between say 3 angstrom and 4 angstrom.

> We have re-drawn the figure as the referee suggests with a surface plot of Sec4 and the Las17 interacting surface shown in a different colour. The legend and text are amended to clarify what is in the figure.

- There are similar issues in supplementary figures with some legends are not understandable (for example residues with shifter changes larger than ??).

> The supplementary figures 3A, 3B and 3C have been re-drawn and text added to legends to clarify

3. There is no evidence showing that the N-terminal three-actin-binding PPR can arrange actin monomers into a trimeric actin seed. If it could, the authors would observe Las17 (300-392) accelerating actin polymerization in the absence of Arp2/3 in the pyrene assay. They would also observe more nucleated actin filaments (instead of or in addition to fast elongating actin filaments) in the TIRF assay.

>We have previously demonstrated that the N-terminal region of the PPR Las17(300-422) which contains the three tracts described here is able to support polymerisation in the absence of Arp2/3 (Allwood et al, 2016 fig6B and 6C). In figure 4, the region used in the pyrene assay is just Las17 amino acids 342-392 which only carries two of the tracts predicted to be involved in this activity. This was a key reason that we looked for other possible actin binding sites that could function alongside the previously recognised sites to support the Arp2/3-independent activity. In terms of the TIRF assays, due to diffraction limits it is not possible to accurately distinguish the signal from a monomer, seed/ nucleus or a very short filament. This approach is not therefore appropriate for assessing stages of nucleation per se which is why we restricted its use to measurement of rate of filament growth.

Minor points:

1. The authors should put forward that the molecular function of the N-terminal PPR is different from that of the C-terminal PPR in the introduction instead of only talking about N-ter PPR.

>we have added text to clarify and have amended Figure 1A to include the N and C terminal region of PPR

2. The naming system is not consistently followed throughout the paper. For instance, mutant names should be italicized, but in lines 360–361, 368, and 396, *las17* P387A is not italicized. In these cases, the *l* is lowercase; while in line 358, the *L* of *Las17* P387A is capitalized and in line 331 *Sla1* P387A. Please review the entire paper to ensure that the yeast naming conventions are consistently applied.

> In *S.cerevisiae* studies it is conventional to indicate genes using italics (uppercase for wild type and lower case for mutants) while for proteins it is convention to use standard format not italics. The convention with regard to proteins as a proper noun (i.e first letter uppercase) is more variable, but a common convention is to use a proper noun format for wild-type protein and all lower case for mutant. We have now checked and applied throughout.

3. When 'actin' is mentioned in the paper, it is not always clear whether it refers to G-actin, F-actin, or both. For example, the MST data shows that *Las17* binds to G-actin, as the experiments were conducted in G-buffer and with low actin concentrations. However, I believe the authors should explicitly state that the experiments were performed with G-actin to ensure that readers without expertise in actin biochemistry can easily understand this point.

> We have amended throughout the main text to clarify.

4. In Fig 1A, the red PP representing Poly Proline is misleading. It looks like two Proline aa. Probably put 'PP' in rectangles to represent motifs?

> The figure has been amended to make the nomenclature clearer. See point 1 above.

5. Fig 3A should definitely go before Fig1A, at least the diagram of the *Las17* domains. Fig 3A contains plenty of information. But the way it presents is very confusing. Why PP motif 1 is coloured in pink and the other two PP motifs are coloured in red? Why 342 and 392 were highlighted in this way? Should PP-N be PPR-N, or N-PPR referring to N-terminal polyproline region?

>We have amended Figure 1A to indicate some of the regions to allow clarity about the domains (as shown point 1 above). We have retained figure 3A however, as aspects of information shown here (eg protein sequence) would be complicated to introduce earlier in the study and are relevant for understanding at this point. We have clarified

legend to explain use of pink and red text.

6. Is there a specific reason to use 0.5 x FME instead of 1 x FME for the pyrene assay? Also, please double check the emission and excitation wavelength of pyrene assay.

>0.5 KME is used as it reduces the rates of actin polymerisation slightly and allows the effect of Las17 to be observed and analysed more readily.

Emission and excitation wavelengths corrected – thank you for noticing

Reviewer #2 (Remarks to the Author):

[Summary]

Hancock et al. investigated the regulation of Las17 by Sla1 and Sec4. Using NMR, pyrene assays, MST, and AlphaFold predictions, they identified a crucial actin-binding domain in Las17 that overlaps with the Sla1-SH3 domains. This finding explains how Sla1-SH3 inhibits Las17-induced polymerization promotion.

The data presented in the manuscript are of good quality and support the authors' findings. While I recommend the study for publication in Communications Biology, the logical progression from data to conclusions is not clearly explained, making the argument difficult to follow. This may be due to some conclusions within paragraphs incorporating findings from later figures. To improve clarity, the authors should carefully explain which aspects of each figure support specific conclusions

>Thank you for reflecting on the areas that were less clear. We have rephrased parts of the text to improve the narrative and support conclusions more effectively.

[Major Comments]

1. Line 173-178: The authors should describe Figures 2B and 2C more carefully to guide readers in understanding, including how the local mobility increase of D52-E55 and E67-V70 have same correlation time are derived.

>Thank you. Additional text has been added to clarify as here

“NMR was used to determine relaxation parameters (^{15}N R_1 , R_2 and NOE) of ^{15}N -labelled SH3#1-2 (Figure 2B). These results show localised increased mobility for the loop around residues D52-E55 in SH3#1 (indicated by the reduced values for R_2 and NOE in the region indicated by a purple bar in Figure 2C) but indicate that residues in the linker between the two SH3 domains (residues E67-V70) have the same correlation time as most other residues in the protein (ie, no reduction in R_2 and NOE in the region indicated by a red bar in Figure 2C), thus demonstrating that the two SH3 domains behave as a single rigid unit.”

2. Line 185-189: The authors should elaborate on the results related to Figures 3B, 3C, and 3D to guide readers in understanding, including why the authors concluded all three PP peptides of Las17 bind to SH3#1 and SH3#2 of Sla1. Providing the K_d values would also be helpful.

>Thank you, another useful comment. We have added extra text to lines 187-193. We have also added the affinities, as suggested, in the figure legend to Figure 3.

“The three peptides bind at similar locations, although peptide 1 binds approximately three times more weakly. The approximate affinities for PP1, PP2 and PP3 obtained from these data are respectively 70, 22 and 24 μM for SH3#1 and 550, 160 and 190 μM for SH3#2.”

3-1. Line 193-196: The description explaining why the 28 aa spacing is the optimal choice, with SH3#1 binding either PP2 or PP3 and SH3#3 binding PP1, is missing.

>A description is in text in line 170-174 a few lines above to that referred to here, however text has been amended to improve clarity.

3-2. Also, why does the affinity of SH3#1&2 become 100 times tighter than SH3#1 alone (Fig. 1C), even though both SH3#1 and SH3#2 bind to PP2 or PP3 without any preference?

>The 100-fold increase in affinity is a manifestation of avidity. The text has been modified to make this point more clearly, and to point the reader more clearly to two relevant references.

4. Description of Figure 3E is missing.

>Added

5-1. Line 231-234: The authors have not fully addressed the point in the following sentence: 'The designation of three distinct actin binding sites (ABS1, ABS2, and ABS3) lying adjacent to the PP motifs 1, 2, and 3, respectively.'

>We have amended text to clarify

"The G-actin binding sites are numbered to align with the PP motifs in Figure 3A, i.e. ABS1 being tested is overlapping with PP1; ABS2 with PP2 and PP3 with ABS3."

5-2. Additionally, the PP motif (the Sla1 binding site) should be included in ABS1, rather than being described as adjacent?

>Text amended (as in point 5-1)

6. Does the model in Figure 5D meet the binding affinity estimated in Figure 4B?

The model in Fig 5D was included to show that it is stereochemically feasible for Las17 to bind to three G-actin monomers and was not intended to convey any implication about affinity. However, the brief discussion of avidity in the text (see point 3-2 discussed above) emphasises that the short length of the linkers between the actin binding sites would be expected to strengthen the affinity considerably. It also highlights similarities to other actin longitudinal nucleators such as Spire which has three actin binding sites at a very similar spacing to those identified here.

7-1. Line 289: Please write down the binding affinities of both reactions, so 40-fold increase is more supported.

Using an Allosteric sigmoidal least squares fit best fit values apparent K_d is $1.388e-006$ for Las17 and Sla1 and $3.418e-008$ for Las17, giving a 40.6-fold difference in K_d .

7-2. Also, what level of competition do the authors anticipate? The authors mention that the ternary complex of Las17: actin: SH3 may form. To support this hypothesis, even when SH3#3 dissociates, the affinity remains at 87 nM (Fig. 1C), which may still be sufficient to form the ternary complex. I believe that a careful explanation of Figure 7A will help clarify the authors' reasoning. The authors may be able to use the data on Sec4-induced inhibition of SH3 binding to Las17 (Fig. 7D) to help explain why ABS1 is the only essential domain for Las17 in the solution assay.

> We have clarified the text as we do not consider that a ternary complex forms. Rather, and as shown in Figure 7A, SH3 domain binding and actin binding is competitive though binding of the multiple SH3 domains of Sla1 is strong due to effects of avidity. We

suggest that factors alleviate the SH3 domain inhibition of Las17 by also binding in the region. This alleviation of inhibition then facilitates actin binding and enhances opportunities for filament polymerisation. One of the factors that we show works in this way is Sec4.

8. What is actin concentration in Figure 7D? Is the assay identical to the other pyrene assays in this manuscript?

>Actin concentration is the same as in other assays (3 μ M). The assays do not always occur at the same rate as different preps of actin can show greater or lesser propensity to polymerise. For this reason we include internal controls.

9. How was the lifetime quantified in Figures 9A, 9B, and 9C? The lifetime of Las17 P387A appears shorter compared to the kymograph in the left panel. It may be helpful to add lines to the kymographs to indicate how the lifetime was determined. Additionally, please include the Y-axis scale in the kymograph.

>Lifetimes are calculated from the time lapse movies not from kymographs themselves as this represents the raw data format. Low and fluctuating signal intensities at the endocytic site for some proteins can mean that the exact start point of recruitment is challenging to determine and t_0 is only recorded from the point that an increase in intensity is consistently recorded. This means that for Las17-GFP where there is a prolonged duration of fluctuating intensity signal at the earliest stages the time the lifetimes measured are more likely to be lower than observable in the kymograph format. Kymographs have been amended to include a size bar to allow a sense of distance of invagination to be gained. However, given this is light microscopy the diffraction limit means that the fluorescence signal is apparently larger than the measurable size of an invagination for example by electron microscopy. It is most appropriate to compare the shape of the kymographs to observe relative differences between behaviours in the wild type and *las17P387A* expressing cells.

Figure 9. The effect of *las17 P387A* mutation on endocytic reporter protein behaviour (A-C) Lifetime of patches from wild type (red) or *las17 P387A* (blue) expressing cells tagged with fluorescent markers, alongside representative kymographs. (A) Las17-GFP, (B) GFP-Abp1, (C) Arc-15 mCherry (describing Arp2/3 complex lifetime). Statistical test Unpaired Mann Whitney test, **** indicates p value < 0.0001. A size bar for each set of kymographs is shown on the upper right of each set. For A and C size bar is 100 nm and for B is 200 nM. (D) Time lapse images showing co-localisation of Las17-GFP (green dots) and Arc15-mCherry (red dots) in wild type cells and *las17 P387A* cells. The initial *las17 P387A* images contain two adjacent endocytic sites. The site of interest is labelled in the first two time panels with a white arrow. Exposure 0.5 sec with 1 second time lapse. 120 seconds recorded. (E) Quantification of lifetime of Las17 alone (green), Arc15 alone (red) and overlap between Las17 and Arc15 (yellow) from multiple endocytic patches is shown (n=13).

[Minor concerns]

Line 92: 'it' appears to be a typo?

We have checked this and the word is relevant for what is being said.

Line 173: 'localized' to be a typo?

word endings are now consistent

Figure S4B legend: 'Actin; 300 μ M' must be a typo.

>Indeed – corrected

Line 433: The phrase 'generate filaments' is not precise. Consider using 'promote actin polymerization' instead.

>Amended

In the Discussion, I respect the authors' passion for their consideration of a set of regulatory streams involving multiple factors. In current manuscript, however, I would suggest focusing solely on the protein-protein interactions and regulatory functions in the molecular level based on the experimental results presented in this manuscript.

Respectfully, we think that it is appropriate to articulate the relevance of the study in this broader context especially because of the high prevalence of both SH3 domains and proline-rich regions in many proteins especially those involved in cell polarity and motility.

COMMSBIO-25-0107-T

We thank the reviewers for their further comments and have incorporated changes accordingly.

Reviewer comments are in black and our responses in red.

Reviewer 1

We quote the entire first and third paragraphs of the referee's response in full because we need to reply to these as a whole.

- I am not convinced (yet) that full length Las can nucleate actin filaments by itself. In the Allwood et al. paper that the authors mentioned, they found 300-422 and 300-536 helps nucleation (Allwood et al. Fig 6C) but it seems the longer the peptides are the capacity of nucleation decrease and capacity of elongation increase. It is still unclear that if the non-truncated Las can help nucleation.

[...]

The authors claim that the TIRF assay cannot be used to observe actin nucleation because short actin filaments cannot be distinguished from a seed or an actin monomer. However, they were performing live imaging, and the filaments would grow, right? Eventually, by examining images taken later, they should be able to determine whether the initial dot was a real filament. In fact, the TIRF assay has been widely used to study the nucleation of actin-related proteins (Graziano et al., MBoC 2011; Sun et al., Nat Comm 2021; Mu et al., NCB 2019 ...).

On receiving this response, we wrote to the Editor for clarification, because the experimental results reported in the paper are focussed on the N-terminal polyproline region of Las17 (residues 300-422) and no results are presented for full length Las17. Actin nucleation by full length Las17 is not stated, nor is it an aim of the work presented in the manuscript. Rather, we report on an investigation into the region of the protein that carries the actin binding sites. We have focussed on this part because we were seeking to understand how it functions given its importance in actin associated events in vitro and in vivo (e.g. Rodal et al, 2003; Feliciano et al, 2015; Sun et al, 2017). The full-length protein has numerous other binding sites including those which can induce multimerization (Feliciano et al, 2012, 2015) inclusion of which would have made outcomes relevant to this study complex to interpret.

The reviewer's comments are concerned with *full length Las17* and *non-truncated Las17*. The Editor then checked with the referee, who replied that they were indeed referring to the constructs used in your study and not the full length Las17.

Throughout the paper, our focus is on the role of Las17(300-422) in actin polymerization, rather than nucleation. For example, the word 'nucleation' appeared once in the Results section, but the word 'polymerization' appeared a total of 21 times. We therefore feel that the reviewer's comments, which are almost all about full length Las17 and nucleation, are outside the subject of this paper. We do not feel that a discussion of the details of previously published work Allwood et al (2016) is of relevance to the arguments raised here.

We do, however, agree that the role of Las17 in nucleation is important, and that quantification of the TIRF result is relevant. Therefore, we revisited our TIRF results and, applying the methodologies suggested by the reviewer, and based on Graziano et al 2011, we can state that Las17 does indeed increase nucleation. This result is now included in the revised version (supplementary figure 4 and in main text lines 275-280).

Specifically, using 500 nM actin \pm 50 nM Las17 at 10 minutes post polymerisation, we measured filament number for actin only at 16.3 filaments \pm 2.3 per unit area, while actin + Las17 gave about triple the number of filaments at 42.3 filaments \pm 7.0.

[second paragraph]

The other issue is that the pyrene assay can only measure actin polymerization and cannot distinguish between nucleation and elongation of actin filaments.

Agreed

The authors claim that Las can induce a signal increase in the pyrene assay.

This is shown in Figure 4 and Figure 7 and also in Urbanek et al, 2013; Allwood et al, 2016 and Tyler et al, 2021.

They also demonstrated that Las can enhance the elongation rate in the TIRF assay. Without clear data ruling out the possibility that the pyrene signal increase is solely due to accelerated actin elongation, I remain unconvinced that full length Las17 can facilitate actin nucleation.

The analysis of the TIRF results stated above now supports a role in actin nucleation. This role is clearly attributable to Las17 as noted above and this is the focus of the work presented. The C-terminal end of Las17 has been studied in detail by us and by others, and clearly binds to actin. However, this work aims to establish the function of the N-terminal polyproline region and the identified additional actin binding sites. We suggest that the referee is here disputing something that is not part of the investigation addressed here.

We **do** show that Las17 can bind 3 actin monomers and we draw comparison with other proteins that bind 3 actin monomers in a longitudinal fashion.

-Fig 5D. This does not constitute scientific results. If the authors want to demonstrate that there is enough space between two actin monomers for the linker, a more scientific approach would be to: (1) measure and provide the distance between two monomers in an actin seed or filament, (2) determine the number of residues present in the linker, (3) assume that the linker is completely flexible, and (4) compare the length of an xx-long peptide with the measured distance to conclude whether there is sufficient space for the linker to fit between the two actin monomers. Even the flexible loop does not fold arbitrarily. There are rules to follow to avoid clashes between residue side chains, to respect backbone dihedral space etc. Therefore, I cannot accept manually placed peptides as a part of main results.

>We take your point, and have done as suggested, and removed the manually placed peptides and replaced with lines.

From clarified comments -

Since your study concludes that "Las17 has the capacity to generate filaments" (lines 565-566), we believe the final points given by this reviewer are sufficiently important and would like to see them being addressed.

>the text referred to here was already changed in the revised version of the paper uploaded in response to a comment by reviewer 2.

Reviewer #2 (Remarks to the Author):

Although some responses still do not fully address my comments, the content is now clearer. I recommend that this study is worthy of publication.

-Line 1142-1143: Add explanation about black and blue lines.

>Added

-Line 1178: 200 nM should be 200 nm (lower case).

>Thank you! Now corrected

-Line 1182-1183: Las17 and Arc15 should be Las17-GFP and Arc15-mCherry, respectively.

>Amended

Response to Reviewers COMMSBIO-25-107B

Our comments in red

REVIEWERS' COMMENTS:

Reviewer #1 (Remarks to the Author):

1. The results with TIRF assays seem convincing.

Thank you for acknowledging

2. Regarding Fig. 5D, I believe it is standard practice in the field to use a straight dashed line to represent regions with no structural information, as I suggested in the first round of revisions. This convention is as intuitive to structural biologists as using dashed lines for bonds and solid lines for clashes.

We have now changed the solid line to a dashed line

The authors discovered that Nter PPR can bind multiple actin monomers and promote actin polymerization. This process is regulated by the binding of the Sla1 SH3 domain, particularly the tandem Sla1 SH3 domains. I appreciate the significance of this work, which may interest a broad community studying the actin cytoskeleton in yeast. However, I believe the authors should make an effort to help readers better understand their data and concepts.

Major points:

1. The authors experimentally demonstrated that three sites in the Las17 PPR contribute to actin binding, and their modelling also supports this hypothesis. However, the experiments were conducted with mammalian actin, while the modelling was performed using yeast actin. It is widely accepted that actin is highly conserved in eukaryotes, and rabbit muscle actin is commonly used for most *in vitro* research. The authors should acknowledge this and discuss the reliability of their data. For instance, they could address whether the residues at the interface predicted by the modelling are conserved between yeast and mammalian actin.
2. The structure figures have significant room for improvement. I advise the authors to read some structural biology papers or/and seek advice from structural biologists on how to properly present 3D protein structures. They can also find useful information on PyMOLwiki.
 - Fig 2A. the interface hydrogen bond distance are hardly readable. Every label should be readable; otherwise, there is no point of putting a label. If the authors wish to show both the overall structure and the interface, I suggest adding a zoomed-in figure in the right panel. Additionally, I think a dark dashed line, rather than bright yellow, would be more readable against the white background.
 - Fig 5A. Not readable. Is it possible to merge A and B by making a zoom in figure?
 - Fig 5B. I have no clue what these yellow dash lines are. Again, labels are not readable. The authors did not give the information regarding R6 and R7.
 - Fig 5C. It is possible to adjust the label placement in PyMOL to prevent it from overlapping with the residues.
 - Fig 5D. Instead of manually placing these selected residues, the authors could simply connect the red residues with a dashed line. This approach is commonly used by structural biologists to indicate regions where structural information is missing. More importantly, it is misleading to position the actin in this way and to suggest that the trimeric actin arrange as actin seed (see Major point 3.)
 - Fig 7E. What are those yellow dashes? This is not informative at all. Instead, the authors can show the surface of the interface. Or they can show the residues with distance between say 3 angstrom and 4 angstrom.

- There are similar issues in supplementary figures with some legends are not understandable (for example residues with shifter changes larger than ??).
3. There is no evidence showing that the N-terminal three-actin-binding PPR can arrange actin monomers into a trimeric actin seed. If it could, the authors would observe Las17 (300-392) accelerating actin polymerization in the absence of Arp2/3 in the pyrene assay. They would also observe more nucleated actin filaments (instead of or in addition to fast elongating actin filaments) in the TIRF assay.

Minor points:

1. The authors should put forward that the molecular function of the N-terminal PPR is different from that of the C-terminal PPR in the introduction instead of only talking about N-ter PPR.
2. The naming system is not consistently followed throughout the paper. For instance, mutant names should be italicized, but in lines 360–361, 368, and 396, *las17* P387A is not italicized. In these cases, the *l* is lowercase; while in line 358, the **L** of Las17 P387A is capitalized and in line 331 **Sla1** P387A. Please review the entire paper to ensure that the yeast naming conventions are consistently applied.
3. When 'actin' is mentioned in the paper, it is not always clear whether it refers to G-actin, F-actin, or both. For example, the MST data shows that Las17 binds to G-actin, as the experiments were conducted in G-buffer and with low actin concentrations. However, I believe the authors should explicitly state that the experiments were performed with G-actin to ensure that readers without expertise in actin biochemistry can easily understand this point.
4. In Fig 1A, the red PP representing Poly Proline is misleading. It looks like two Proline aa. Probably put 'PP' in rectangles to represent motifs?
5. **Fig 3A should definitely go before Fig1A**, at least the diagram of the Las17 domains. Fig 3A contains plenty of information. But the way it presents is very confusing. Why PP motif 1 is coloured in pink and the other two PP motifs are coloured in red? Why 342 and 392 were highlighted in this way? Should PP-N be PPR-N, or N-PPR referring to N-terminal polyproline region?
6. Is there a specific reason to use 0.5 x FME instead of 1 x FME for the pyrene assay? Also, please double check the emission and excitation wavelength of pyrene assay.